# PROTPAINTER: DRAW OR DRAG PROTEIN VIA TOPOLOGY-GUIDED DIFFUSION

**Zhengxi Lu** [*][◇]  **Shizhuo Cheng** [*][◇]  **Yuru Jiang** [◇]  **Yan Zhang** [†][◇]  **Min Zhang** [†][◇]

◇Zhejiang University

{3200105645,12218138,3220102689,zhang_yan,min_zhang}@zju.edu.cn

## ABSTRACT

Recent advances in protein backbone generation have achieved promising results under structural, functional, or physical constraints. However, existing methods lack the flexibility for precise topology control, limiting navigation of the backbone space. We present **ProtPainter**, a diffusion-based approach for generating protein backbones conditioned on 3D curves. ProtPainter follows a two-stage process: curve-based sketching and sketch-guided backbone generation. For the first stage, we propose **CurveEncoder**, which predicts secondary structure annotations from a curve to parametrize sketch generation. For the second stage, the sketch guides the generative process in Denoising Diffusion Probabilistic Modeling (DDPM) to generate backbones. During this process, we further introduce a fusion scheduling scheme, Helix-Gating, to control the scaling factors. To evaluate, we propose the first benchmark for topology-conditioned protein generation, introducing Protein Restoration Task and a new metric, self-consistency Topology Fitness (scTF). Experiments demonstrate ProtPainter's ability to generate topology-fit (scTF > 0.8) and designable (scTM > 0.5) backbones, with drawing and dragging tasks showcasing its flexibility and versatility.

## 1 INTRODUCTION

Recently, generative models based on the denoising diffusion framework (Ho et al., 2020; Song et al., 2020) have shown remarkable success in creating realistic protein backbones, including RFD-iffusion (Watson et al., 2023), Genie (Lin & Alquraishi, 2023), FrameDiff (Yim et al., 2023b) and others. Flow-based models like FrameFlow (Yim et al., 2023a) and FOLDFLOW-OT (Bose et al., 2023) incorporate techniques such as flow matching or Riemannian optimal transport, demonstrating unprecedented performance in designability and efficiency (Zheng et al., 2024). A follow-up question is how users could harness their imagination to guide the generation process in producing proteins with the desired structure and functions, expanding the possibilities beyond what is achievable with those unconditional generation models.

Some models explored non-structural conditions such as biochemical properties (Hsu et al., 2024) and desired functions (Komorowska et al., 2024; Kulytė et al., 2024). Others attempted to generate backbones with structural conditioning like contact maps (Harteveld et al., 2023), partial structural motifs (Watson et al., 2023; Lin et al., 2024; Ingraham et al., 2023), symmetry (Ingraham et al., 2023; Watson et al., 2023), and point cloud (Ingraham et al., 2023; Long et al., 2022). The above methods provide various forms of structural constraints designed for downstream tasks like motif-scaffolding, binder design, and symmetric oligomers design. However, they lack a conditioning mechanism for more detailed and precise structural control, thus posing a challenge for delicate backbone space navigation, which is crucial for tasks like multi-state design (Praetorius et al., 2023) and allosteric protein design (Pillai et al.).

Topology with secondary structure (also referred to as **Blueprint** or **Coarse-Grained Topology, CG Topo**) (Correia, 2024; Harteveld et al., 2023; Zhang et al.; Harteveld et al., 2022; Huddy et al., 2024; Huang et al., 2014; 2022) is a more programmable mechanism. It represents proteins at a higher level

---

[*]Equal contribution.

[†]Corresponding author.

of abstraction, focusing on the arrangement and connectivity of secondary structure elements such as $\alpha$-helices and $\beta$-strands, allowing designers to specify the overall shape and structural organization of a protein. This mechanism is particularly useful for creating proteins with repetitive structural features, such as multi-subunit protein assemblies (Lutz et al., 2023) and nanomaterial (Huddy et al., 2024). However, current methods are heavily based on linear, parametric topology configuration, which defines the topology with a set of parameters that describe the geometric relationships between Secondary Structure Elements (SSE). While powerful, it suffers from limited coverage of the three-dimensional topology space, potentially constraining the diversity of protein designs that can be generated, not to mention editing the topology of proteins flexibly. Expanding beyond these parametric constraints could open up possibilities for exploring a broader range of protein topologies and enable more controllable protein design.

**Our approach**   We present the first method that uses 3D curves as topological constraints to define protein folds. The 3D curve representation encompasses critical structural features, including the number of helices, their relative lengths, orientations, positions, and the curvature of helices, providing a more detailed and precise topology description.

In analogy to super-resolution tasks in imaging (Choi et al., 2021), our objective is equal to refine the coarse-grained 3D curves into fully structured protein backbones. In this context, our curve condition corresponds to the reference image $y$, and the generated backbones correspond to the refined images in the DDPM framework. For imaging, reference and generation can be easily aligned and sampled into a latent space of the same dimensions. But for protein structures, the alignment could be hard. To address this challenge, we propose a CurveEncoder that upsamples the condition curve into a sketch, while a filtering operation downsamples the generated backbones into a frame. This ensures that the condition curve and the backbone frames share the same dimensionality. To enhance the quality of translation generation, we build on the approach from RFDiffusion (Watson et al., 2023), utilizing RoseTTAFold estimates $pX_0$ for translational guidance based on self-conditioning (Chen et al., 2022). In summary, our method introduces a novel approach that leverages naive sketches to bridge curves and backbones, extending the DDPM framework to enable more flexible topology control.

**Main Contributions**   The main contributions are summarized as follows:

1. We propose **ProtPainter**, the first method to generate protein backbones with specific topology based on 3D curves. For the first stage of ProtPainter, sketches are parametrically created with the assistance of a CurveEncoder, which extracts local geometric features and predicts the SSE of curves. For the second stage, we present a retraining-free method to guide the generative process in DDPM and generate designable backbones based on a given reference sketch. During this process, a sketch fusion scheduling mechanism, Helix-Gating, is used to determine the scaling factor by incorporating helix-percentage guidance.

2. We provide a benchmark to evaluate curve or topology-conditioned backbone generation. 1) We propose a new metric scTF to assess the topological similarity between the generated backbone and curve condition. 2) We offer a method to generate a dataset of curves from protein backbones and implement a series of curve-based operations, including jointing, dragging, and drawing. And 3) we also develop a Protein Restoration Task to compose the benchmark.

3. We demonstrate that this new modality can be well applied to downstream tasks such as binder design, motif scaffolding, and dragging. Notably, an empirical example demonstrates that for multi-state designs like hinge proteins, ProtPainter provides an easy way to create preliminary scaffolds with good quality for further design.

## 2   BACKGROUND AND RELATED WORK

### 2.1   DIFFUSION PROBABILISTIC MODELING

The Diffusion Probabilistic Modeling (Sohl-Dickstein et al., 2015) formulates the model training as follows. Given a forward diffusion process, the model predicts the noise added to the original sample at time $t$. For a sample from the training set $x_0$, the forward process is defined as iteratively

adding a small amount of Gaussian noise to the sample in $T$ steps, which produces a sequence of noisy samples $x_{0:T}$ such that the final sample $x_T \sim \mathcal{N}(0,1)$ to a good approximation. Within the framework of Denoising Diffusion Probabilistic Modeling (DDPM) (Ho et al., 2020), the noise magnitude at each step is defined by a variance schedule $\beta_t, t \in [0:T]$ such that

$$p_t(x_t|x_{t-1}) = \mathcal{N}(x_t, \sqrt{1-\beta_t}x_{t-1}, \beta_t I) \tag{1}$$

The above transition defines a Markov process in which the original data is transformed into a standard normal distribution. It is possible to write the density of $x_t$ given $x_0$ in a closed form as

$$p_t(x_t|x_0) = \mathcal{N}(x_t, \sqrt{\bar{\alpha}_t}x_0, (1-\bar{\alpha}_t)I), \quad s.t. \quad x_t = \sqrt{\bar{\alpha}_t}x_0 + \sqrt{1-\bar{\alpha}_t}\epsilon_t, \tag{2}$$

where $\bar{\alpha}_t = \Pi_i^t \alpha_i$ and $\alpha_i = 1 - \beta_i$ and $\epsilon_t \sim \mathcal{N}(0,1)$.

Transforming a sample $x_T$ into sample $x_0$ is done in several updates that reverse the process of adding destructive noising, given by a reverse sampling scheme

$$x_{t-1} = \frac{1}{\sqrt{\alpha_t}} \left( x_t - \frac{\sqrt{1-\alpha_t}}{1-\bar{\alpha}_t}\epsilon_\theta(x_t, t) \right) + (1-\alpha_t)\epsilon, \tag{3}$$

where $\epsilon \sim \mathcal{N}(0,1)$. The neural network $\epsilon_\theta$ (the denoiser) is trained to predict noise added to $x_0$.

## 2.2 Guided Diffusion for Backbone Generation.

Guided sampling has been applied in diffusion models to generate samples with human instructions, formulating the sampling process as conditional and unconditional terms. Considering conditioning variable $y$, classifier-guided methods like Chroma (Ingraham et al., 2023), train a time-dependent classifier model $p_t(y|x)$ on the noised structures $x_t \sim p_t(x|x_0)$ and adjust the sampling posterior $\nabla_x \log p_t(x|y)$ by Bayesian inference. Instead of training a classifier to estimate $p_t(y|x)$, classifier-free methods approximate the conditional term heuristically. Wang et al. (2024) and Komorowska et al. (2024) introduce the physical force to extend an unconditional model to dynamic conformation sampling. Force guidance is also applied to generate antibody (Kulytė et al., 2024) with lower energy.

## 2.3 Topology-based Backbone Control

Controlling protein topology has been a long-standing challenge to go beyond the linear configuration of SSE (also known as **Blueprint** (Xu & Zhang, 2018). Common blueprint-based methods describe the number and approximate locations as well as the overall orientation of the secondary structure elements parametrically (Kortemme, 2024; Harteveld et al., 2022; 2023; Westhead et al., 1999), which is preliminary but essential for *de novo* protein design (Kortemme, 2024), binder design (Levy et al., 2004), and membrane protein design (von Heijne, 2006). These parameterized representations are powerful for repetitive assembly but limited by their degree of freedom, toughening general users to control more complex 3D topology flexibly.

**Topology-based Diffusion Models for Backbone Generation** To gain more control like traditional blueprint-based methods, some diffusion-based models have taken topology as a condition: Topodiff (Zhang et al.) encodes the global topology in the latent space with VAE, enabling topological control by giving a querying protein, but not supporting more detailed topology editing. DiffTopo (Correia, 2024) employs the diffusion model to generate the sketch from a predefined SSE sequence and then inputs it to RFDiffusion for a realistic backbone. However, its topology definition follows a linear configuration and lacks control of three-dimensional topology. In conclusion, current topology-based diffusion models do not support more flexible and detailed topology control.

## 3 Method

The generation process is split into two stages: curve sketching (introduced in Section 3.1) and sketch-guided sampling (introduced in Section 3.2). To improve the guided sampling process, Helix-Gating, a fusion scheduling mechanism, is detailed in Section 3.3.

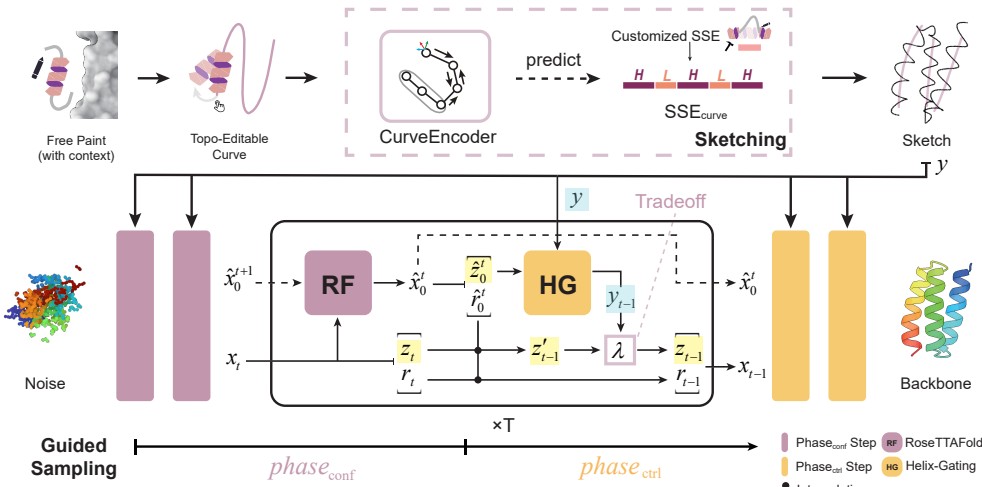

Figure 1: Architecture. **Sketching**: given a 3D curve, SSE$_{\text{curve}}$ is predicted by CurveEncoder. Then a naive sketch is generated parametrically. **Guided Sampling**: the sketch is fused into a diffusion sampling process with the guidance of RoseTTAFold and Helix-Gating interpolation.

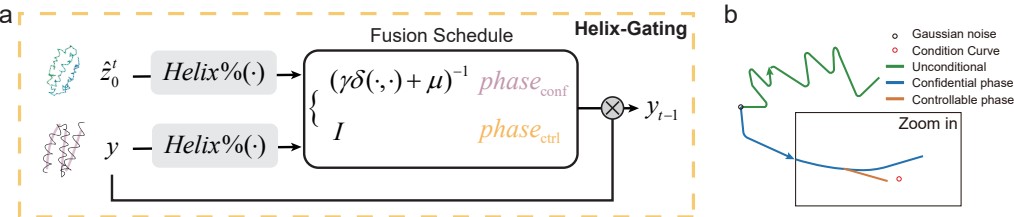

Figure 2: Sketch Fusion Scheduling with Helix-Gating. a. Helix-Gating splits the sampling process into two phases by comparing the helix percentage of $\hat{z}_0^t$ and $y$, enabling the scheduling of fusion. b. The curve space trajectories of different diffusion sampling processes.

### 3.1 SKETCHING CURVES

This stage begins by defining a forward process that abstracts backbone topology into curve representations. Next, we introduce the CurveEncoder, which annotates curves with SSE labels, denoted as SSEcurve. The final sketch is then generated parametrically, guided by SSEcurve.

**Topology Represented by Curves** To represent the topology of protein $C_\alpha$ backbones as curves, we employ a downsampling method detailed in Appendix A.1. Specifically, $\alpha$-helices and $\beta$-sheets are abstracted to their central axes, while loop regions retain their original coordinates. The resulting curve coordinates are re-sampled, smoothed, and annotated with SSE labels by averaging the labels of their nearest backbone atoms.

**CurveEncoder** This module is designed to predict the SSE annotation for curves SSE$_{\text{curve}}$, as a reverse process for curve SSE assignment. Inspired by Greener & Jamali (2022), a three-layer EGNN (Satorras et al., 2021) is applied to extract connectivity features of curve coordinates, and a one-dimension CNN to extract the curvature feature as a complement. Then a multi-head attention layer integrates the features and predicts the secondary structure element annotation for SSE$_{\text{curve}}$. SSE$_{\text{curve}}$ can be customized by user input. Given node embeddings $h^l$ and coordinate embeddings $x^l$ of layer $l$, and edge information $\varepsilon = (e_{ij})$, the Equivariant Graph Convolutional Layer (EGCL) is written as $h_{l+1}, x_{l+1} = EGCL[h_l, x_l, \varepsilon]$. The algorithm is shown in Algorithm 1, and details are shown in Appendix D. The parametric approach to generate the naive sketch with curve coordinates and SSE$_{\text{curve}}$ is shown in Appendix E.

### 3.2 SKETCH-GUIDED BACKBONE SAMPLING

Our model uses the frame representation following (Watson et al., 2023), which comprises the translation $z$ ($C\alpha$ coordinates) and rotation $r$ ($N$-$C\alpha$-$C$ rigid orientation) for each residue. Consider $X_t = [z_t, r_t]$ are the residue frames at diffusion step $t$, where $z_t \in \mathbb{R}^{N_{res} \times 3}$ are the coordinates of $C\alpha$ (translation part) and $r_t \in SO(3)^{N_{res}}$ is the rotation matrix (rotation part).

The translation part is generated from the 3D Gaussian noise by DDPM.

$$p(z_{t-1}|z_t, z_0) = \mathcal{N}(z_{t-1}; \tilde{\mu}(z_t, z_0), \tilde{\beta}_t I_3), \quad \text{with} \quad \tilde{\beta}_t = \frac{1 - \overline{\alpha}_{t-1}}{1 - \overline{\alpha}_t}\beta_t \approx \beta_t, \tag{4}$$

where

$$\tilde{\mu}(z_t, z_0) = \frac{\sqrt{\overline{\alpha}_{t-1}}\beta_t}{1 - \overline{\alpha}_t}z_0 + \frac{\sqrt{\alpha_t}(1 - \overline{\alpha}_{t-1})}{1 - \overline{\alpha}_t}z_t, \tag{5}$$

and $z_0$ can be estimated by RoseTTAFold prediction $\hat{z}_0$ with masked sequence input, inspired by RFDiffusion.

For residue orientations, Brownian motion is used on the manifold of rotation matrices. The frames are equivariant with respect to rotation.

$$p_\theta(x_{t-1}|x_t) = p_\theta(R * x_{t-1}|R * x_t), \quad \text{where} \quad R * x_t = [Rz, Rr]. \tag{6}$$

Given a reference naive sketch $y \in \mathbb{R}^{N_{res} \times 3}$, we define a conditional distribution of guidance term $y_{t-1}$:

$$y_{t-1} \sim q(y_{t-1} \mid y, \hat{z}_0^t). \tag{7}$$

Choi et al. (2021) refines the generation x conditioned on the reference y by

$$p_\theta(x_{t-1}|x_t, y) \approx p_\theta(x_{t-1}|x_t, \phi(x_{t-1}) = \phi(y_{t-1})). \tag{8}$$

where $\phi$ is a linear low-pass filtering operation to ensure the low-pass features of the reference image and the generation images remain the same. We adopt this idea of aligning generation to reference at low dimensions with sketch being the bridge. The proposed CurveEncoder upsamples the condition curve to a sketch and $\phi$ filters the generation backbones into a frame. The generated backbone frames $x_0$ can now be guided by reference sketch $y$ through the filtered frame part $\phi_\lambda(x_0)$. Here we define the frame filter operation as $\phi_\lambda(X_t) = \lambda z(X_t)$ where $z(X_t)$ extracts the $C\alpha$ coordinates of frames $X_t$ and $\lambda$ (between 0 and 1) is a factor for tradeoff between diversity and guidance.

We approximately treat rotation ($r_t$) and translation ($z_t$) as independently distributed variables under a general translation condition $c_T$.

$$p_\theta(x_{t-1}|x_t, c_T) = p_\theta(z_{t-1}|x_t, c_T)p_\theta(r_{t-1}|x_t) \quad \text{if} \quad p_\theta(r_{t-1}|x_t) = p_\theta(r_{t-1}|x_t, c_T). \tag{9}$$

Combining equations 8 and 9, we have

$$p_\theta(x_{t-1}|x_t, y) \approx p_\theta(z_{t-1}|x_t, \phi(x_{t-1}) = \phi(y_{t-1}))p_\theta(r_{t-1}|x_t). \tag{10}$$

We only need to update $z_{t-1}$

$$z_{t-1} = \phi(y_{t-1}) + Iz'_{t-1} - \phi(x'_{t-1}) \tag{11}$$

where $x'_{t-1}$ is sampled from the unconditional distribution proposed by $x_t$, $x'_{t-1} \sim p_\theta(x'_{t-1}|x_t)$ and $z'_{t-1}$ is the translation part of $x'_{t-1}$. Set operation $\phi$ on the equation 5, we have

$$\phi(z'_{t-1}) = \frac{\sqrt{\alpha_t}(1 - \overline{\alpha}_{t-1})}{1 - \overline{\alpha}_t} \cdot \phi(z_t). \tag{12}$$

So the conditional probability approximation is

$$z_{t-1} = \frac{\sqrt{\overline{\alpha}_{t-1}}\beta_t}{1 - \overline{\alpha}_t} \cdot \hat{z}_0^t + \frac{\sqrt{\alpha_t}(1 - \overline{\alpha}_{t-1})}{1 - \overline{\alpha}_t} \cdot (1 - \lambda) \cdot z'_t + \lambda \cdot y_{t-1}. \tag{13}$$

### 3.3 Helix-Gating: Controlling Scaling Factors

We propose **Helix-Gating**, a two-stage fusion scheduling scheme to enhance the guided sampling process with sketch $y$. The transition timing between (1) the confidential phase and (2) the controllable phase is determined by comparing the helix percentage (operator denoted as $\mathcal{O}$) between RF predicted $\hat{z}_0^t$ and sketch $y$. In the confidential phase, the guidance is limited and scaled using the difference in helix percentages between predicted and target proteins, ensuring a constant fidelity increase with limited guidance from sketch $y$. In the controllable phase, the guidance is fully provided:

$$y_{t-1} = \frac{\sqrt{\alpha_t}(1 - \overline{\alpha}_{t-1})}{1 - \overline{\alpha}_t} \cdot \mathcal{F}(y, \hat{z}_0^t) \cdot y \tag{14}$$

$$\mathcal{F}(y, \hat{z}_0^t) = \begin{cases} \gamma \cdot \delta(\mathcal{O}(\hat{z}_0^t), \mathcal{O}(y)) + \eta, \text{ if } \mathcal{O}(\hat{z}_0^t) < \mathcal{O}(y) & \text{(Confidential Phase)} \\ I, & \text{if } \mathcal{O}(\hat{z}_0^t) \geq \mathcal{O}(y) & \text{(Controllable Phase)} \end{cases} \tag{15}$$

where $\gamma$ and $\eta$ are hyper-parameters, $\delta$ is the difference function. We set $\lambda = 2/3$, $\gamma = 0.2$, $\eta = 0.7$, according to the ablation study G.5. Figure 2.b illustrates this process: once the helix percentage reaches a threshold, the controllable phase begins, and more condition information is integrated, refining the generation to align closely with the sketch. Ablation study 5 and Appendix 12.c demonstrate the effectiveness of Helix-Gating.

### 3.4 Drawing Binder and Dragging protein

Dragging a protein is formulated as curve-conditioned motif scaffolding. In this process, the curve of the dragged protein serves as the scaffold to be generated, conditioned on its updated shape, while the fixed part is treated as the motif condition. For a structure with length $L$, let $M$ and $S$ be the index set of motif and scaffold, respectively, that is $M \cup S = \{1, ..., L\}$. So let the structure of motif and scaffold be $x^M$ and $x^S$, respectively. The whole un-noised structure is $x_0 = [x_0^M, x_0^S]$. The $x_0^M$ backbone and sidechain structure are input as fixed templates of RoseTTAFold to predict $\hat{z}_0^S$, which influences the translation part, so with the motif fixed or masked, the denoising process becomes $p_\theta(x_{t-1}^S | x_t^S, x_0^M, c_T)$. We approximate that

$$p_\theta(x_{t-1}^S | x_t^S, x_0^M, c_T) \approx p_\theta(z_{t-1}^S | x_t^S, c_T) p_\theta(r_{t-1}^S | x_t^S, x_0^M) \tag{16}$$

To draw binders, we first calculate the distance between the curve and the target protein to identify "interface hotspots" $z_h$ on the target. The complex design is then conditioned on both the curve and the target protein, incorporating these hotspot residues.

## 4 Experiments

### 4.1 Protein Restoration

We evaluate the conditioning effectiveness and generation quality of ProtPainter by our proposed Protein Restoration task, which is defined as given the condition representing an existing protein target, a deep learning method generates designable backbones to fit the curve topologically (details in Figure 7). We conduct experiments in two aspects: (1) preliminary topology fitness before refolding[1] and (2) topology fitness and designability after refolding with comparison. Considering the curve as a novel condition, the goal is to demonstrate that our method achieves more precise and fine-grained control over topology while maintaining high generation quality.

**Metrics** We evaluate the restoration capability of methods mainly on the three metrics:

- **Designability**. Designability is quantified through backbone TM-score before and after refolding (*scTM*, higher is better). In this work, each curve generates $N_{bb}$ backbones and refolding pipeline uses ProteinMPNN (Dauparas et al., 2022) at temperature 0.1 to generate $N_{seq}$ sequences for Omegafold (Wu et al., 2022) to predict;

---

[1]Design sequence and predict all-atom structure.

- **Confidence**. Confidence in refolding is measured as the *pLDDT* of the predicted structures to test the Local-Distance Difference;
- **Similarity**. To measure this, we propose *self-consistency Topology Fitness (scTF)*, which is calculated as the Procrustes similarity between the refolded (or un-refolded) backbone curve and the curve guidance. The threshold for scTF is chosen from the experience of topological similarity (shown in Figure 6) and the proportional relationship with scTM (shown in Figure 13).

**Preliminary Topology Fitness** For preliminary evaluation of the topology control ability, we choose representatives of six topology families, three architectures in Mainly Alpha class from CATH (Orengo et al., 1997; Sillitoe et al., 2021)[2]. In this experiment, ProtPainter attempts to restore them conditioned on their curve representations (no refolding needed). In Table 1, we find that ProtPainter is capable of generating structures barely similar (scTF > 0.7) to more than 60 percent of diverse topologies in Mainly Alpha class (CATH ID = 1). And it performs well especially on topologies in the Up-down Bundle (CATH ID = 1.20) with the portion of scTF > 0.7 near 0.8. However, the portion goes down to 0.249 for DNA polymerase (CATH ID = 1.10.150) for its topology complexity.

Table 1: Protein Restoration Task on CATH without refolding.

| CATH ID | 1.10.150 | 1.10.287 | 1.20.5 | 1.20.58 | 1.20.120 | 1.25.40 | 1 |
|---|---|---|---|---|---|---|---|
| # count | 369 | 976 | 299 | 428 | 738 | 570 | 1082 |
| scTF > 0.7 | 0.249 | 0.642 | 0.849 | 0.797 | 0.816 | 0.675 | 0.615 |
| scTF > 0.8 | 0.0759 | 0.470 | 0.716 | 0.633 | 0.626 | 0.375 | 0.394 |

**Topology Fitness with Designability** To evaluate the topology fitness and designability, we select 10 monomer structures as cases. These cases are restored by ProtPainter conditioned on curves and refolded to evaluate their designability (shown in Table 2). We can restore most topologies and generate both similar and designable structures but there are some exceptions like 1AV1, 7KUW, and 4DB8. We also display some cases in Figure 3, including binder design and motif scaffolding.

Table 2: Cases of Protein Restoration Task with refolding.

| ID | 1P68 | 6S9L | 1TQG | 7KUW | 4DB8 | 2N8I | 1TJL | 1AV1 | O14842 | P30968 |
|---|---|---|---|---|---|---|---|---|---|---|
| length | 82 | 249 | 93 | 55 | 220 | 84 | 118 | 205 | 284 | 319 |
| scTM | 0.9655 | 0.9093 | 0.9588 | 0.5149 | 0.7318 | 0.7071 | 0.6862 | 0.3292 | 0.9508 | 0.9396 |
| scTF | 0.944 | 0.956 | 0.980 | 0.854 | 0.771 | 0.628 | 0.614 | 0.920 | 0.938 | 0.942 |
| pLDDT | 94.883 | 86.986 | 86.788 | 48.34 | 80.809 | 85.826 | 79.044 | 88.566 | 94.754 | 86.894 |

**Baselines** We compare our method with state-of-the-art unconditional diffusion model RFDiffusion (Watson et al., 2023) and conditional diffusion models Chroma (Ingraham et al., 2023), and TopoDiff (Zhang et al.). The modalities of conditions are as follows,

- **RFDiffusion** is conditioned on sequence lengths.
- **Chroma** is conditioned on secondary structure annotations.
- **Chroma** incorporates point clouds as the condition, instead of secondary structure annotations. These point clouds are generated by constructing a curved cylinder using a 3D curve as the central axis, with a radius varying from 0 to 4 Å. The point cloud is uniformly distributed within the cylinder's volume.
- **TopoDiff** is conditioned on the topology latent, such as latent-based linear interpolation between two structures, which is the most relevant to our method. To compare it with ours, curves are preprocessed by our *CurveEncoder* and transformed into sketches. Then the sketches are mapped into the latent space as TopoDiff's DDIM conditions.

---

[2]CATH ID is composed of **C**lass, **A**rchitecture, **T**opology and **H**omologous Superfamily which are separated by ".".

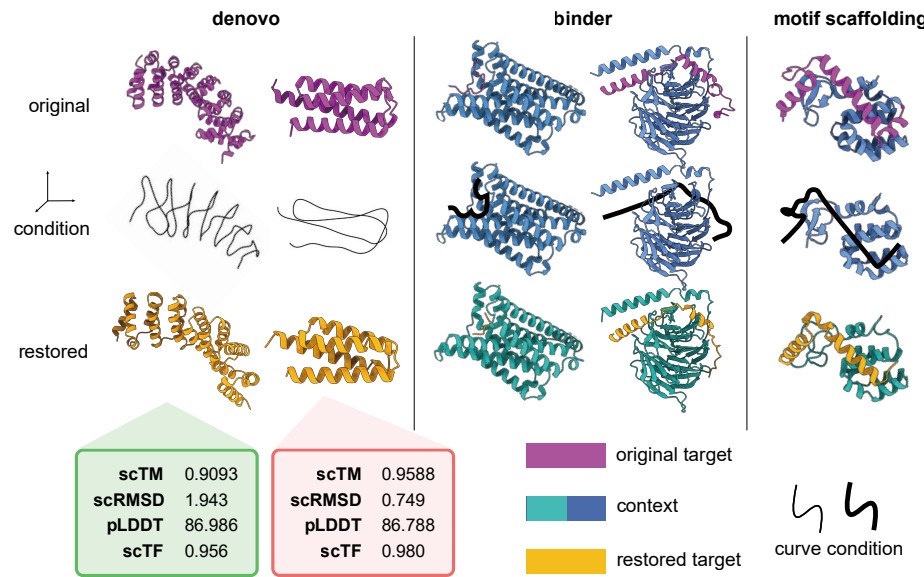

Figure 3: Examples of ProtPainter on *de novo* protein design, binder design, and motif scaffolding. From left to right, the original structures are 6s9l, 1tqg, 7f4d_MR, 7f4d_GB, and 103l. Curves are visualized in 3D space. Other examples are shown in Figure 14.

**Data** For the evaluation dataset, We select three representative protein clusters ordered by increasing length and topological complexity: HHH_ems[3], 1a0b_cluster[4], and GPCR[5]. Each comprises 50 backbones to restore (refold needed). Proteins in the same dataset share similar topologies but exhibit subtle structural differences (as shown in Figure 8.a).

Table 3: Comparison of Similarity and Designability on Protein Restoration Task. Confident designability (**CD**) is the portion of proteins with scTM > 0.5 and pLDDT > 70, and fit designability (**FD**) is the portion of proteins with scTM > 0.5, pLDDT > 70, and scTF > 0.7. Each metric is evaluated with 500 backbones selected from 4000 sequences

| Condition | Method | HHH_ems | | | | | med | | | | | GPCR | | | | |
|---|---|---|---|---|---|---|---|---|---|---|---|---|---|---|---|---|
| | | scRMSD↓ | scTM↑ | scTF↑ | CD↑ | FD↑ | scRMSD | scTM | scTF | CD | FD | scRMSD | scTM | scTF | CD | FD |
| - | RFDiffusion | **0.753** | **0.910** | 0.693 | **0.980** | 0.640 | 1.064 | **0.962** | 0.272 | 0.974 | 0.050 | 2.402 | **0.905** | 0.262 | 0.886 | 0.000 |
| SSE | Chroma | 3.284 | 0.812 | 0.382 | 0.868 | 0.066 | 5.001 | 0.834 | 0.227 | 0.782 | 0.002 | 9.022 | 0.776 | 0.182 | 0.658 | 0.000 |
| PointCloud | Chroma(r=0) | 3.837 | 0.742 | 0.414 | 0.812 | 0.088 | 8.411 | 0.658 | 0.251 | 0.464 | 0.044 | 18.855 | 0.486 | 0.260 | 0.208 | 0.000 |
| | Chroma(r=1) | 4.011 | 0.737 | 0.342 | 0.784 | 0.144 | 10.390 | 0.630 | 0.259 | 0.544 | 0.042 | 20.405 | 0.474 | 0.229 | 0.182 | 0.000 |
| | Chroma(r=2) | 3.068 | 0.744 | 0.290 | 0.862 | 0.026 | 8.754 | 0.660 | 0.239 | 0.640 | 0.000 | 17.157 | 0.512 | 0.198 | 0.220 | 0.000 |
| | Chroma(r=3) | 2.496 | 0.798 | 0.340 | 0.880 | 0.028 | 8.816 | 0.677 | 0.238 | 0.560 | 0.002 | 17.497 | 0.509 | 0.201 | 0.202 | 0.000 |
| | Chroma(r=4) | 4.048 | 0.727 | 0.374 | 0.840 | 0.000 | 7.138 | 0.717 | 0.215 | 0.582 | 0.000 | 17.368 | 0.477 | 0.227 | 0.166 | 0.000 |
| Topology | TopoDiff | 0.871 | 0.897 | 0.350 | 0.958 | 0.160 | 1.762 | 0.911 | 0.247 | **0.992** | 0.020 | 11.420 | 0.653 | 0.179 | 0.340 | 0.000 |
| Curve | ProtPainter(Ours) | 2.635 | 0.718 | **0.767** | 0.832 | **0.654** | 4.926 | 0.763 | **0.791** | 0.870 | **0.734** | 9.431 | 0.892 | **0.800** | **0.936** | **0.792** |

**Comparison results** The results are shown in Table 3. First, compared to state-of-the-art methods, ProtPainter demonstrates convincing performance in confident designability (CD), reflecting its capability to generate high-confidence designs. Second, ProtPainter excels in understanding human-defined topologies and achieves state-of-the-art control over them, as evidenced by its superior values and trends in FD and scTF metrics. In contrast, other conditional methods are limited

---

[3]A cluster of simple proteins consisting of 3-helix bundles connected by loops, with lengths between 50 and 60

[4]Top 50 proteins in PDB (Burley et al., 2023; Berman et al., 2003) structurally similar to 1a0b.pdb using PDBefold (Dietmann et al., 2001), with lengths ranging from 100 to 250 and no redundant structures

[5]The largest superfamily of cell surface membrane receptors, encoded by approximately 1000 genes, characterized by conserved seven-transmembrane (7TM) helices connected by three intra- and three extracellular loops (Rosenbaum et al., 2009; Eichel & von Zastrow, 2018; Katritch et al., 2012; Zhang et al., 2024), with lengths between 280 and 400

in the precise topological control. Point clouds utilized by Chroma only offer vague spatial conditioning and lack sufficient internal topological information for precise control. Moreover, protein latent space used by TopoDiff reveals an ambiguity in understanding sketches, thus struggling to restore the overall topology defined as human-drawn curves. Finally, ProtPainter slightly outperforms RFDiffusion in FD for shorter proteins (length $< 60$), likely due to the simplicity and similarity of protein folds at this length. In summary, ProtPainter offers significantly more precise and detailed topological control over the backbone while maintaining high design quality.

## 4.2 USER STUDY: CURVE FROM SCRATCH

In addition to generating sketches from existing protein structures, we introduce three approaches enabling users to create novel 3D curves:

1. Users can modify an existing 3D curve by dragging anchor points to reshape it, generating a new curve.

2. A 2D curve can be drawn from scratch on a sketchpad for convenience. This curve is then converted into 3D by assigning random, smoothly varying depths to its points. Experimental results are presented in Table 4.

3. By installing a ChimeraX plugin, users can draw curves directly on a protein surface, defining binder conditions. Secondary structure elements for the generated binder can also be assigned. The plugin installation code is available at `https://github.com/lll6gg/ChimeraX_plugin_binder`.

For Method 2, we begin by generating 100 two-dimensional curves from scratch. Among these, 80 are created using a 2D curve generator, while the remaining 20 are drawn manually by humans. Depth values are then assigned to the points, varying randomly yet smoothly. The low-frequency component is modeled as a sinusoidal function, with random noise added as high-frequency variations. Notably, all curves have lengths of less than 100, and each contains fewer than six helix bundles. Some of the cases are shown in Figure 15 .

Table 4: Results conditioned on curves from scratch.

| Source | scRMSD↓ | scTM↑ | scTF↑ | CD | FD |
|---|---|---|---|---|---|
| Human | 3.278 | 0.782 | 0.745 | 0.75 | 0.65 |
| Curve Generator | 3.565 | 0.768 | 0.696 | 0.8875 | 0.575 |
| All | 3.508 | 0.771 | 0.706 | 0.86 | 0.59 |

Additionally, to simulate the user drawing process, we introduced noise by randomly perturbing each point within a sphere centered at its original position. The results in I demonstrate our ability to understand and address potential challenges faced by users.

## 4.3 NAVIGATION OF PROTEIN TOPOLOGY SPACE WITH PROTPAINTER

To demonstrate that curves as the prerequisite condition enable flexible topology editing while retaining designability, we show empirical cases such as dragging, rotating, joining, and local SSE editing. Novelty is calculated as the maximum TM score (*maxPDBTM*) relative to PDB (Burley et al., 2023; Berman et al., 2003) with Foldseek (Van Kempen et al., 2024). As shown in Figure 4, dragging proteins enables topology transition from the three-helix bundle (2) to (3) and finally to the two-helix scaffold (4), all retaining designable (scTM $> 0.5$). Local SSE editing enables drastic backbone change while the topology remains unchanged (TF $= 0.983$). A comprehensive example of hinge protein (Praetorius et al., 2023) design (6,7,8) demonstrates that ProtPainter-generated helix bundles can transit between different topologies while retaining designability. The topology space trajectory of the jointing domains (9, 10) at different angles (points in purple) indicates that the novel topology (11) can be obtained in this way.

In conclusion, with the help of topo-editable curves, ProtPainter achieves unprecedented structural control in diffusion-based backbone generation, enabling more natural, flexible, and precise topology space navigation compared to traditional structure editing.

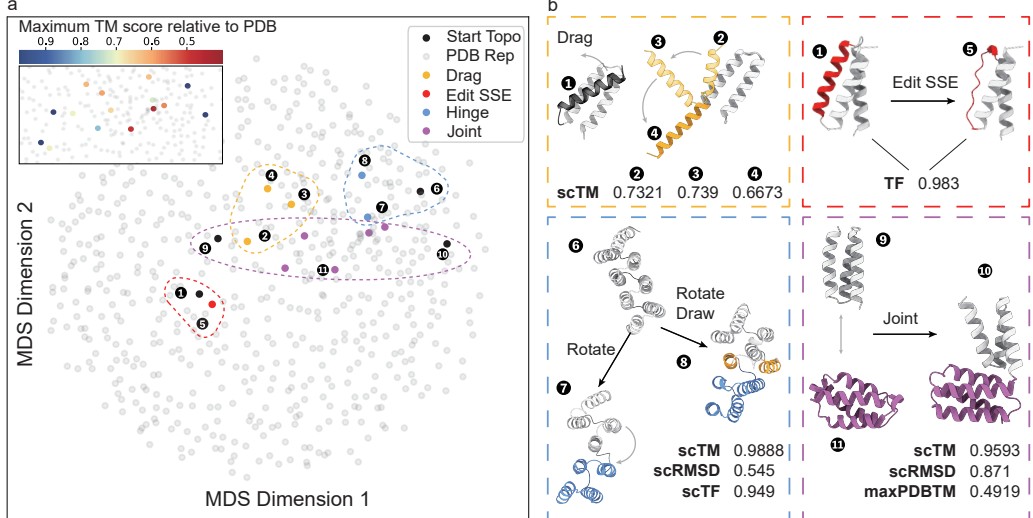

Figure 4: Draw and edit process. (a) Structures are visualized in the MDS topology space, with their colors corresponding to respective operations. Novelty is measured as the maximum TM score relative to PDB (Burley et al., 2023; Berman et al., 2003) in the upper left corner. (b) Case study for actions of dragging, SSE editing, comprehensive tasks like hinge protein, and jointing.

## 4.4 ABLATION STUDY

We perform experiments on key components of ProtPainter. The parameters are set as $N_{\text{curve}} = 50$ per dataset, $N_{\text{bb}} = 5$, and $N_{\text{seq}} = 8$. We define fit-designability as scTM > 0.5, pLDDT > 70, and scTF > 0.8. The results are shown in Table 12. We conduct the ex If no Helix-Gating scheme is applied, the Two-Phase transition occurs at timestep = 10. If no CurveEncoder is used, secondary structure elements (SSE) are randomly labeled based on the target portion. Additional ablations are provided in Appendix G.

Table 5: ProtPainter ablations.

| Fit-designability (↑) | | Curve Encoder | Helix- Guiding | Two- Phase | Helix- Gating |
|---|---|---|---|---|---|
| GPCR | HHH_ems | | | | |
| 0.676 | 0.56 | ✓ | ✓ | ✓ | ✓ |
| 0.656 | 0.53 | ✓ | | ✓ | ✓ |
| 0.588 | 0.48 | ✓ | ✓ | ✓ | |
| 0.668 | 0.392 | ✓ | ✓ | | |
| 0.388 | 0.20 | ✓ | | | |
| 0.112 | 0.196 | | ✓ | ✓ | ✓ |
| 0.296 | 0.18 | | ✓ | | |

## 5 CONCLUSION

In this work, we propose ProtPainter, which applies a new representation of structural condition – curves – to generate designable protein backbones with topological control, enabling powerful topology space navigation through curve-based operations, enriching current protein design methods. We already tested our method on many downstream tasks with amazing results. Considering the sampling time, the sketching process is efficient, but backbone generation and refolding are time-consuming, taking between 10 seconds and 2 minutes on a single NVIDIA. To enable real-time protein design, more optimizations are needed to reduce inference time. Furthermore, we plan to work on beta sheet in future work, which is briefly discussed in Appendix K.

## 6 ACKNOWLEDGMENTS

This work is supported in part by the National Science and Technology Major Project Grant No. 2022ZD0117000 to M.Z., the National Nature Science Foundation of China Grant No. 62202426 to M.Z., the National Natural Science Foundation of China Grant No. 92353303/32141004/81922071 to Y.Z., and the 'Pioneer' and 'Leading Goose' R&D Program of Zhejiang Grant No. 2024C03147 to Y.Z..

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

# A    TOPOLOGY FITNESS

## A.1    CURVE AS TOPOLOGY REPRESENTATION

Given the $C\alpha$ coordinates $\mathbf{X}$ of a protein, we define its topology by abstracting its secondary structure elements. For the alpha helix and beta sheet, the $C\alpha$ coordinates are replaced by a series of points along the central axis of that segment. Loop regions retain their original $C\alpha$ coordinates. A resampling procedure is then applied to reduce the number of coordinates and average the point distance along the curve, yielding the curve coordinates $\mathbf{C}$. The curve coordinates are then categorized as a secondary structure by averaging the nearest neighboring $C\alpha$ secondary structure, resulting in a final curve representation $\mathbf{Y}$.

## A.2    DEFINITION OF TOPOLOGY FITNESS

To compare the topological structures of two proteins, we introduce *Topology Fitness (TF)*, a coarser measure than TM-score. First, the down-sampling transformation is applied to both proteins, generating simplified 3D curves. The curves are then sampled to have the same number of points and aligned using Procrustes analysis. It involves normalizing two curves to achieve optimal overlap by minimizing the Procrustes distance, which quantifies the difference in shape between objects. Using Procrustes superposition, an approach that translates, rotates, and scales the objects, the position and size of the curves can be adjusted to maximize their alignment. This process ensures that the curves are as close as possible in both position and scale. Finally, disparsity is calculated, representing the degree of alignment, based on the sum of distances between corresponding points on the two curves(Peres-Neto & Jackson, 2001; Zhao et al., 2019).

Mathematically, given two coordinate sets $\mathbf{Y}_1 = \{y_{1,1}, y_{1,2}, \ldots, y_{1,m}\}$ and $\mathbf{Y}_2 = \{y_{2,1}, y_{2,2}, \ldots, y_{2,m}\}$, disparsity is computed via Procrustes analysis:

$$\text{disparsity} = \min_{R} \|\mathbf{Y}_1 - \mathbf{Y}_2 R\|_F,$$

where $R$ is an orthogonal matrix aligning $\mathbf{Y}_2$ to $\mathbf{Y}_1$, and $\|\cdot\|_F$ denotes the Frobenius norm. TF is written as

$$\text{TF} = 1 - \text{disparsity}.$$

## A.3 SCTF WITH SCTM AND SCRMSD

The scTF is defined as self-consistency Topology Fitness, measuring the topology fitness between structures and condition curves. Here scTF is computed as the TF between designed structures (after refolding) and conditions (scTF_2 in Figure 7). From the joint distribution of Figure 5, scTF has a close linear relationship with scTM and scRMSD, with a positive correlation with the former and a negative correlation with the latter. scTF can also be used to measure the designability of a curve (whether it describes the topology of a designable backbone). scTF's value changes more in line with scTM as its length changes, because it is more similar to scTM, paying more attention to global similarity, is less sensitive to length, and is not as strict as scRMSD in long structures.

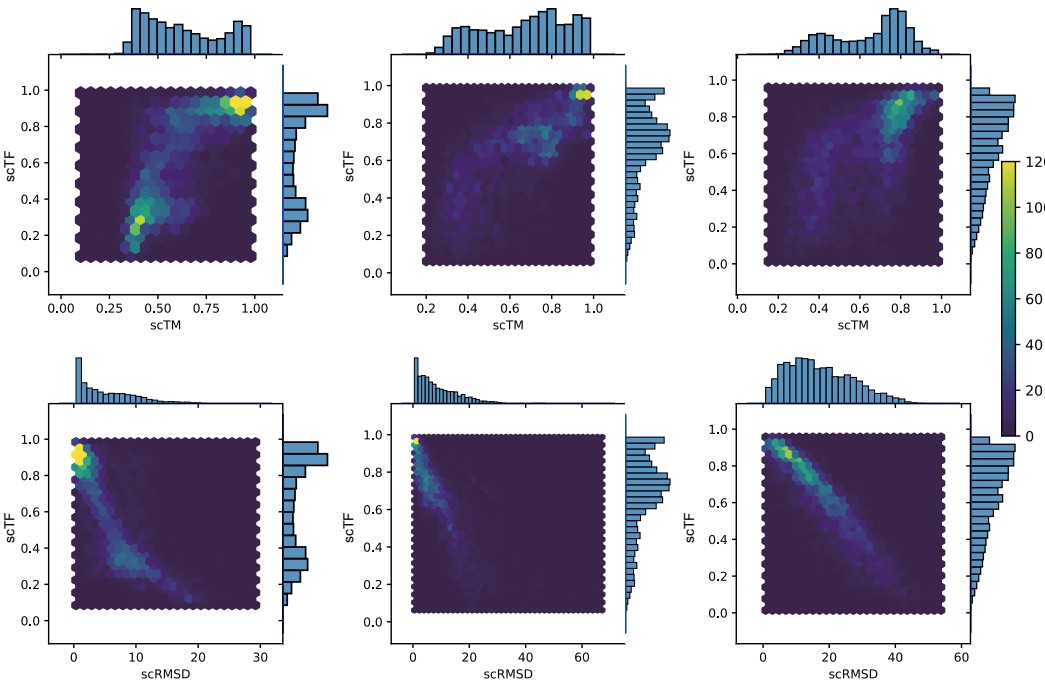

Figure 5: scTF vs scTM and scRMSD. Figures from left to right show the test results on datasets HHH_ems, med, and GPCR respectively. The first and second rows show the relationship between scTM and scTF, and the relationship between scRMSD and scTF, respectively. The results have not been filtered by selection. $N_{curve} = 50, N_{bb} = 10, N_{seq} = 8$.

## A.4 CUTOFF VISUALIZATION

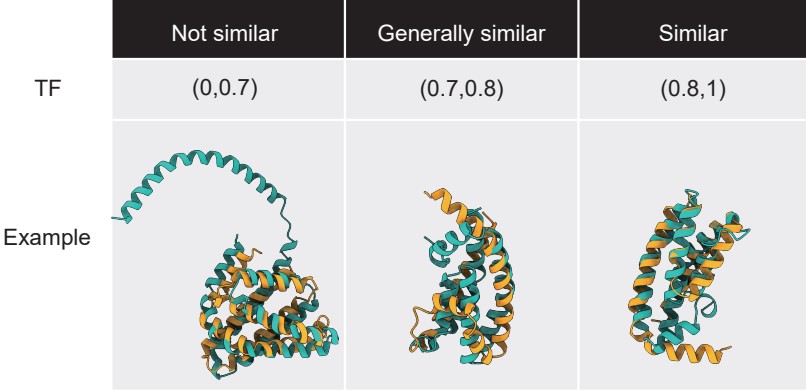

Figure 6: Topology similarity example between two different structures (green and orange).

# B CURVE-PROTEIN RESTORE TASK

## B.1 PIPELINE

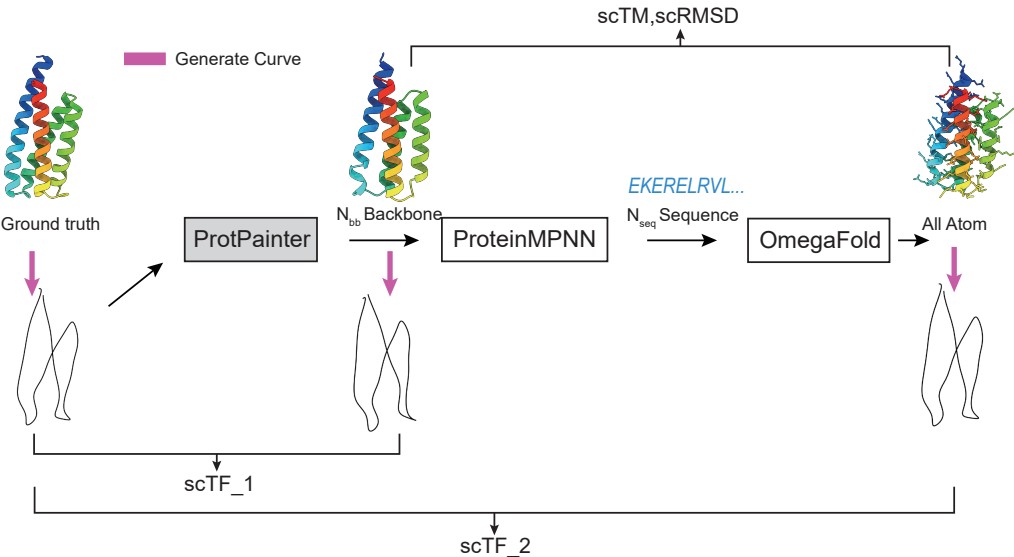

Figure 7: Protein Restoration Task and designability test. Using ProtPainter, we sample $N_{bb}$ backbones conditioned on a curve sampled from ground truth. Then we proceed to sample multiple ($N_{seq}$) sequences with ProteinMPNN (Dauparas et al., 2022). Each sequence is then folded with OmegaFold (Wu et al., 2022) to obtain the predicted backbone, which is scored against the sampled backbone with RMSD (scRMSD) or TM-score (scTM). This framework also gives a method for evaluating topology similarity between structures using scTF. scTF_1 and scTF_2 represent the degree of topology agreement between the generated(before refolding) or designed(after refolding) structures and the condition curve. They have a certain relationship but also distinction. scTF_2 is always larger than scTF_1. We use scTF_1 in Table 1 and scTF_2 in other places.

## B.2 DATASET VISUALIZATION

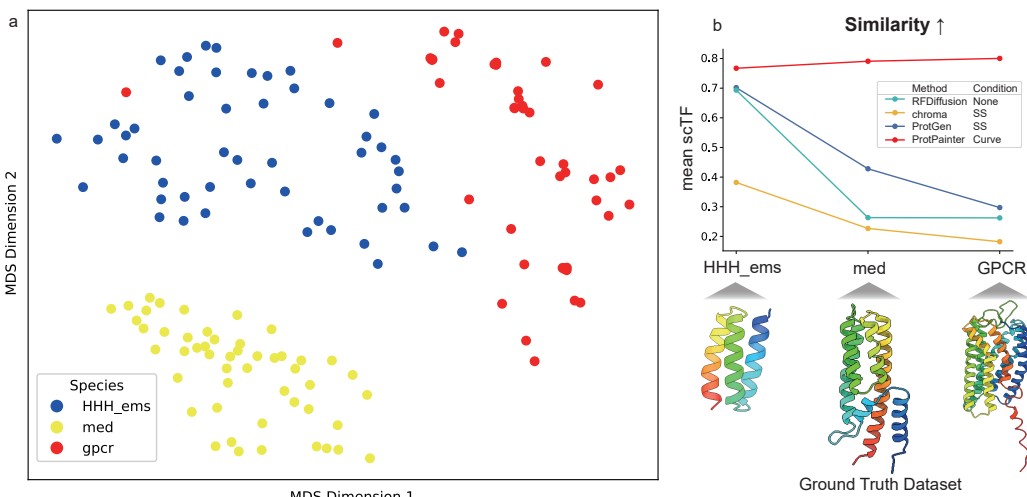

Figure 8: a. Datasets visualization on protein space. We calculate scTM between structures and plot the MDS plot. b. The mean scTF in Figure 3.

## C CURVE SPACE VISUALIZATION

We visualize the generation results compared to RFDiffusion and ProtPainter on curve space to demonstrate that we exert more detailed control.

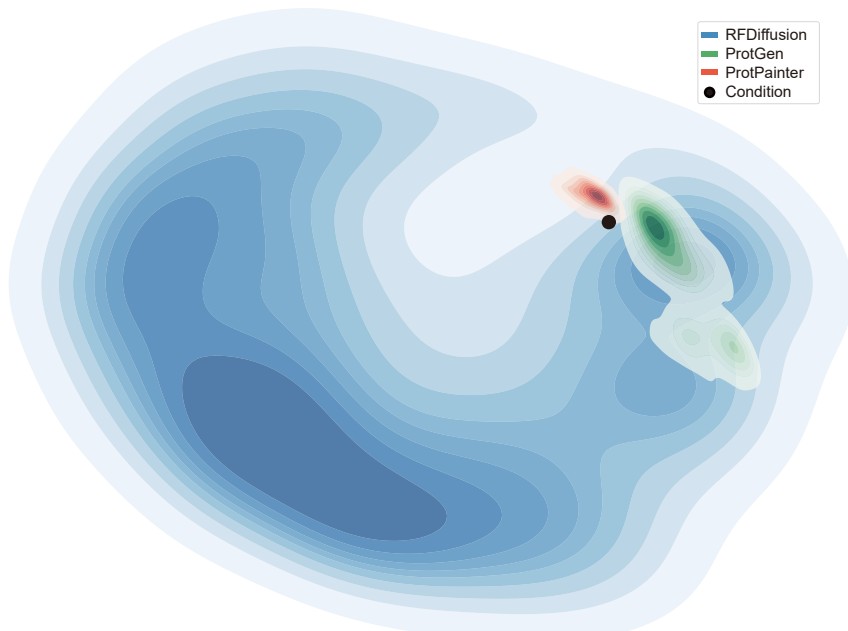

Figure 9: Conditional Generation Visualization. We sample 1500 backbones conditioned on 1tqg for each method. RFDiffusion is conditioned on sequence length. Protein Generator is conditioned on secondary structure lists. ProtPainter is conditioned on the curve. The generated backbones are shown on curve space with MDS measured by TF to each other.

# D    CURVEENCODER

The CurveEncoder aims to extract spatial geometric features from protein topologies to predict the SSE for consequent sketching and aligning.

## D.1    CURVATURE

This process starts with curve interpolation to form a chain, effectively describing the shape of curves in computational geometry while preserving local properties. Protein curves are interpolated using splines, providing a smooth and continuous representation of the protein backbone.

To capture the geometric characteristics, the curvature is defined along the interpolated curves. Curvature quantifies how sharply a curve bends at a given point, offering insight into the protein's local geometric features. Curvature $\kappa(t)$ is calculated as:

$$\kappa(t) = \frac{\|\mathbf{r}'(t) \times \mathbf{r}''(t)\|}{\|\mathbf{r}'(t)\|^3},$$

where $\mathbf{r}'(t)$ and $\mathbf{r}''(t)$ are the first and second derivatives of the spline curve with respect to the parameter $t$, respectively. This formulation captures both the magnitude and direction of bending at each point along the curve.

## D.2    ARCHITECTURE

Building on this geometric representation, we utilize a combination of EGNN (Equivariant Graph Neural Network) and a curvature-based CNN. For the 3D structural data, each node in the 3D graph has scalar features and contains 3D coordinates. Equivariant graph neural networks are proposed to incorporate geometric symmetry into model building (Han et al., 2022). Set Equivariant Graph Convolutional Layer (EGCL) incorporates node embeddings $h^l = \{h_0^l, ..., h_{M-1}^l\}$, coordinate embeddings $x^l = \{x_0^l, ..., x_{M-1}^l\}$, edge information $\varepsilon = (e_{ij})$, $h_{l+1}, x_{l+1} = EGCL[h_l, x_l, \varepsilon]$. The message passing and node updates are

$$m_{ij} = \phi_e\left(h_i^l, h_j^l, \|x_i^l - x_j^l\|^2, a_{ij}\right) \tag{17}$$

$$x_i^{l+1} = x_i^l + C \sum_{j \neq i} \left(x_i^l - x_j^l\right) \phi_x(m_{ij}) \tag{18}$$

$$m_i = \sum_{j \neq i} m_{ij} \tag{19}$$

$$h_i^{l+1} = \phi_h(h_i^l, m_i) \tag{20}$$

where $m_{ij}$ are vector messages. $\phi_e$, $\phi_x$, and $\phi_h$ are functions commonly approximated by Multilayer Perceptrons (MLPs) for edge or node operations. Edge values $a_{ij} = e_{ij}$ without additional edge information. Compared with traditional 3D CNNs, geometrically equivariant GNNs do not require voxelization of input data while still maintaining the desirable equivariance.

The model workflow is as follows:

## D.3    ABLATION

In order to verify the accuracy and generalization of the model prediction, we processed 1000 protein structures in the HHH_ems dataset to generate two different granularity datasets with (curve, SSE) pairs. The two datasets are detailed and have no details, where the number of curve points is 40% and 120% of the $C\alpha$ atomic coordinates, respectively. We trained and evaluated models 1 to 5 in Table 6 on these two datasets. Training is done in 100 epochs with cross-entropy loss, Adam optimization, and a learning rate of 0.01. Results are shown in Figure 10. Method 1, a curvature CNN, performs the best in accuracy on the training dataset, but its limited ability to extract spatial information from its curvature results in poor generalization. Method 5, combining the advantages of curvature CNN and EGNN, performs well in both accuracy and generalization. And we select Method 5 in our approach.

---

**Algorithm 1** CurveEncoder

---

1: **Input:** Curve coordinates $x = \{x_0, \ldots, x_{M-1}\}$
2: $x', t \leftarrow \text{InterpolationSample}(x)$          ▷ Lengths of $x'$ and $t$ are $N$ $(N > M)$
3: $\kappa(t) \leftarrow \frac{\|\mathbf{r}'(t) \times \mathbf{r}''(t)\|}{\|\mathbf{r}'(t)\|^3}$             ▷ Curvature with length $N$
4: $e_{ij} \leftarrow \begin{cases} 1 & \text{if } |i - j| = 1 \\ 0 & \text{otherwise} \end{cases}$          ▷ for $0 \le i, j < N$
5: $h_3^1 \leftarrow \text{EGCL}_i(h_{i-1}^1, x', e), \quad \text{for } i = 1 \text{ to } 3, \ h_0^1 = \kappa(t)$
6: $h^2 \leftarrow \text{CNN}(\kappa(t))$
7: $o \leftarrow \text{Linear}(\text{MultiHeadAttention}(h_3^1, h^2))$
8: **Output:** $o$                ▷ Prediction of Secondary Structure

---

Table 6: CurveEncoders for Ablation

| Method | Models |
|--------|--------|
| 1 | Curvature CNN with kernel 15 |
| 2 | GCN connected by chain with 3D-coord feature |
| 3 | EGNN with random feature |
| 4 | EGNN with curvature feature |
| 5 | EGNN with curvature feature + curvature CNN |

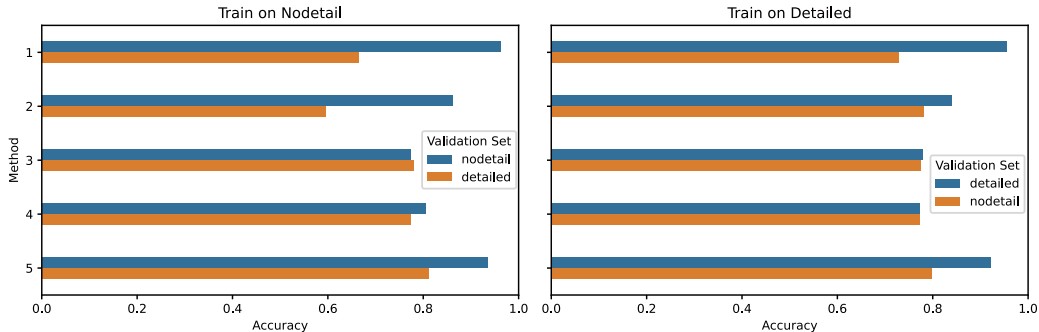

Figure 10: CurveEncoder Ablation Result.

### D.4 TRAIN

We processed 15000 data samples consisting of PDB (Burley et al., 2023; Berman et al., 2003) and scaffolds, generating three different granularity curve datasets with sampling rates of 40%, 80%, and 120%. Then the curves are smoothly interpolated and the true labels are masked partially and randomly to enhance the robustness. Then the dataset is split into a training set and test set in the ratio of 8:2. For EGNN, we set num_tokens to 100, dim to 32, and depth to 3. For training, we set the learning rate at 0.0001, batch size at 1, and run 2000 epochs with CrossEntropyLoss, Adam optimization, and a learning rate of 0.0001.

## E SKETCHING

Inspired by (Harteveld et al., 2022), we place predicted SSEs at their respective relative positions as specified by a given curve, creating a 3D backbone object containing only the SSEs, which we refer to as a naive sketch. For $\alpha$-helix, we generate the sketch coordinates near the curve such that every 3.6 amino acid residues make one turn of the helix, and the upward translation is 0.54nm. Hence, the pitch is 0.54nm, and the distance between two amino acid residues is 0.15nm. We sample coordinates on the curve with some distance for loop and $\beta$-sheet. The naive sketch contains

topological information and more detailed structural information, such as curve curvature, and $\alpha$-helical packing information. The naive sketch is aimed to align the curve prompts and generate backbones. Then it is plugged into conditional generation to guide the design.

## F COMPARISON

The comparison to RFDiffusion in designability and scTM is shown in Figure 11.

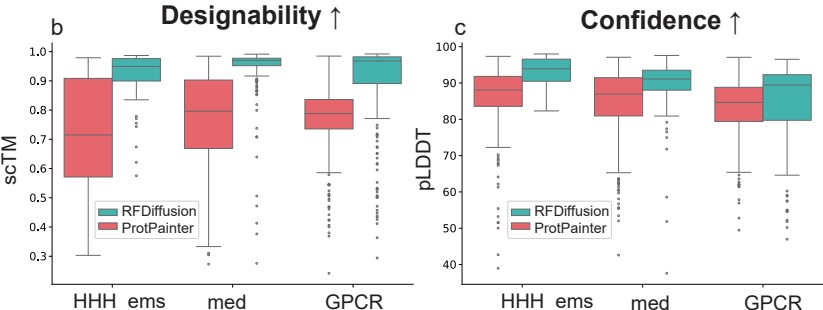

Figure 11: Designability and scTM compared to RFDiffusion.

## G OTHER ABLATION

### G.1 CONDITION

The condition ablation on protein restore task similarity is shown in Figure 7.

Table 7: Condition ablation on protein restore task similarity. Metrics are measured without refolding. *curve with SSE means that users set the ground truth SSE on curves. scTF is scTF_1.

| Dataset | Method | Condition | SSE_percent (↑) | scTF (↑) | RMSD (↓) |
|---------|--------|-----------|-----------------|----------|----------|
| HHH_ems | RFDiffusion | seqlen | 0.905±0.059 | 0.364±0.164 | 10.453±3.708 |
| | ProtPainter | curve | 0.948±0.045 | 0.813±0.115 | **6.416±0.853** |
| | ProtPainter | curve with SSE * | **0.952±0.045** | **0.845±0.109** | 6.686±0.979 |
| med | RFDiffusion | seqlen | 0.771±0.088 | 0.196±0.092 | 16.077±2.068 |
| | ProtPainter | curve | 0.837±0.118 | **0.798±0.203** | **11.174±3.456** |
| | ProtPainter | curve with SSE * | **0.851±0.121** | 0.749±0.217 | 13.811±3.767 |
| GPCR | RFDiffusion | seqlen | 0.778±0.105 | 0.122±0.034 | 24.746±1.175 |
| | ProtPainter | curve | 0.761±0.080 | **0.892±0.075** | **16.633±5.516** |
| | ProtPainter | curve with SSE * | **0.794±0.073** | 0.817±0.104 | 22.597±4.251 |

## G.2 USER CONTROLLABILITY TRADEOFF RESULTS

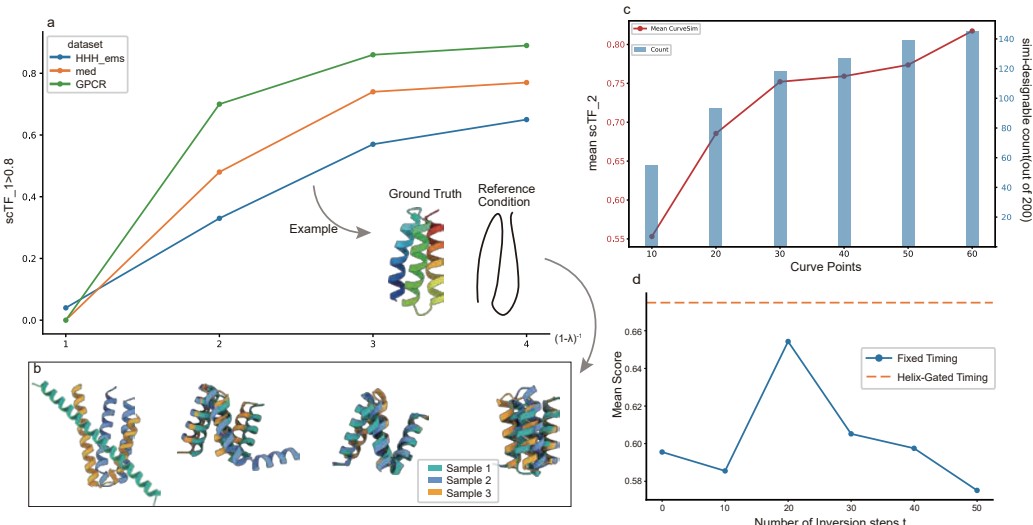

Figure 12: Ablation. a. $\lambda$ control as a tradeoff between diversity and similarity. b. Example samples of different $\lambda$. c. Selection method ablation; Score is computed as (scTF + scTM)/2. d. Two-Phase timing ablation.

Users can control the generation between diversity and similarity mainly by adjusting the down-sampling factor $\lambda$ between 0 and 1 (not included). If $\lambda = 0$, then it becomes an unconditional generation process. In Figure 12.a, we set $N_{curve} = 50/dataset$, $N_{bb} = 5$, $N_{seq} = 8$ and sample $\lambda = 0, 1/2, 2/3, 3/4$, also written by $(1 - \lambda)^{-1} = 1, 2, 3, 4$, generate backbones conditioned on curves of three protein datasets and evaluate by scTF (scTF_1 in Figure 7 without refolding) between the generated backbones and curves. We find that on all three curve datasets formed from existing protein datasets, the similarity increases and diversity decreases when $\lambda$ increases. When $\lambda = 0$, the designs similar to the condition curve are almost 0, but dataset HHH_ems is an exception for the folds of short proteins (50-60) are limited. And we set $\lambda = 2/3$ in the following experiments. Figure 12.b shows the cases how the results vary by $\lambda$.

## G.3 CURVE POINTS

In this study, we sample a protein with an amino acid length of 54 using point counts of 10, 20, 30, 40, 50, and 60 to represent its topological structure. These representations are then utilized as conditions for ProtPainter generation. Our findings reveal that as the conditions become more detailed, the topological similarity to the original protein increases, with a notable plateau observed between 30 and 40 points. Given this trend, we adopt a sampling rate of 40%, which effectively hints at the protein's topology while optimizing the number of representations required.

## G.4 TWO-PHASE TIMING

We conduct an ablation study to elucidate the impact of varying $t$ (i.e., the number of inversion steps) during the Conditional Diffusion steps on datasets HHH_ems ($N_{curve} = 50$, $N_{bb} = 2$, $N_{seq} = 8$). We run our approach on two phases split by fixed timing steps ($t = 0, 10, 20, 30, 40, 50$) or varying helix-gated timing to obtain results (sampling from $t = 50$ corresponds to the pure noisy latent to $t = 0$ corresponds to the denoised results). We evaluate the result by computing the mean average score ($0.5 \times$ scTM + $0.5 \times$ scTF) of the generated backbones. The results are shown in Figure 12.d.

## G.5 HYPERPARAMETER

Table 8: Hyperparameter Ablation. $N_{\text{curve}} = 50$ and $N_{\text{bb}} = 2$ for Protein Restoration Task on dataset HHH_ems with selection method by score. scTF_1 and scTF_2 are shown in Figure 7, average = (scTF_1 + scTM + scTF_2)/3, and CD is the confident designability (scTM > 0.5 and pLDDT > 70).

| Params | scTF_1(↑) | scTM(↑) | scTF_2(↑) | average(↑) | CD(↑) |
|---|---|---|---|---|---|
| $\gamma$=0, $\eta$=0.6 | 0.805 | 0.746 | 0.765 | 0.772 | 0.58 |
| $\gamma$=0, $\eta$=0.7 | 0.806 | 0.716 | 0.775 | 0.766 | 0.54 |
| $\gamma$=0, $\eta$=0.8 | 0.800 | **0.733** | 0.783 | 0.772 | 0.59 |
| $\gamma$=0, $\eta$=0.9 | 0.795 | 0.705 | 0.739 | 0.746 | 0.49 |
| $\gamma$=0, $\eta$=1 | 0.796 | 0.730 | 0.783 | 0.770 | 0.54 |
| $\gamma$=0.1, $\eta$=0.6 | 0.814 | 0.720 | 0.777 | 0.770 | 0.58 |
| $\gamma$=0.1, $\eta$=0.7 | 0.801 | 0.711 | 0.753 | 0.755 | 0.50 |
| $\gamma$=0.1, $\eta$=0.8 | 0.807 | 0.725 | 0.765 | 0.765 | 0.53 |
| $\gamma$=0.1, $\eta$=0.9 | 0.793 | 0.721 | 0.749 | 0.754 | 0.52 |
| $\gamma$=0.1, $\eta$=1 | **0.819** | 0.720 | 0.768 | 0.769 | 0.55 |
| $\gamma$=0.2, $\eta$=0.6 | 0.814 | 0.736 | 0.788 | 0.779 | 0.60 |
| $\gamma$=0.2, $\eta$=0.7 | 0.811 | 0.731 | **0.804** | **0.782** | 0.61 |
| $\gamma$=0.2, $\eta$=0.8 | 0.809 | 0.707 | 0.746 | 0.754 | 0.49 |
| $\gamma$=0.2, $\eta$=0.9 | 0.808 | 0.721 | 0.775 | 0.768 | 0.58 |
| $\gamma$=0.2, $\eta$=1 | 0.796 | 0.696 | 0.748 | 0.747 | 0.49 |
| $\gamma$=0.3, $\eta$=0.6 | 0.803 | 0.745 | 0.779 | 0.774 | **0.62** |
| $\gamma$=0.3, $\eta$=0.7 | 0.816 | 0.727 | 0.757 | 0.766 | 0.51 |
| $\gamma$=0.3, $\eta$=0.8 | 0.802 | 0.736 | 0.766 | 0.768 | 0.61 |
| $\gamma$=0.3, $\eta$=0.9 | 0.801 | 0.718 | 0.753 | 0.757 | 0.53 |
| $\gamma$=0.3, $\eta$=1 | 0.798 | 0.676 | 0.734 | 0.736 | 0.48 |

With $\lambda = 3$, we evaluate hyper parameters $\gamma$ ranging from 0 to 0.3 and $\eta$ from 0.6 to 1, as shown in Table G.5. We select $\gamma = 0.2$ and $\eta = 0.7$ to balance between designability and restoration similarity.

## G.6 SELECT METHOD

We test the selected method of selecting designed structures by score or scTM. We find that selecting by score is more suitable in our task, which seeks a compromise between controllability and designability. From the edge distribution of the results and the case in Figure 13, corresponding to the proportion of scTM>0.5, we chose a threshold of scTF>0.8 to illustrate the similarity between a structure and a condition.

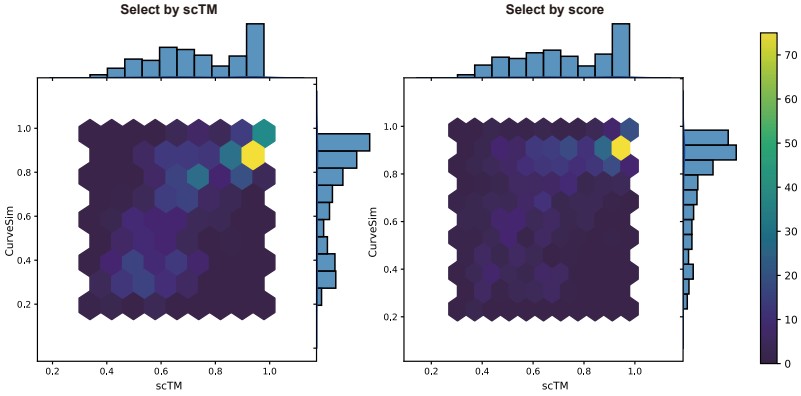

Figure 13: Selection Method Ablation. The score is computed as (scTF+scTM)/2.

## H EXAMPLE GALLERY

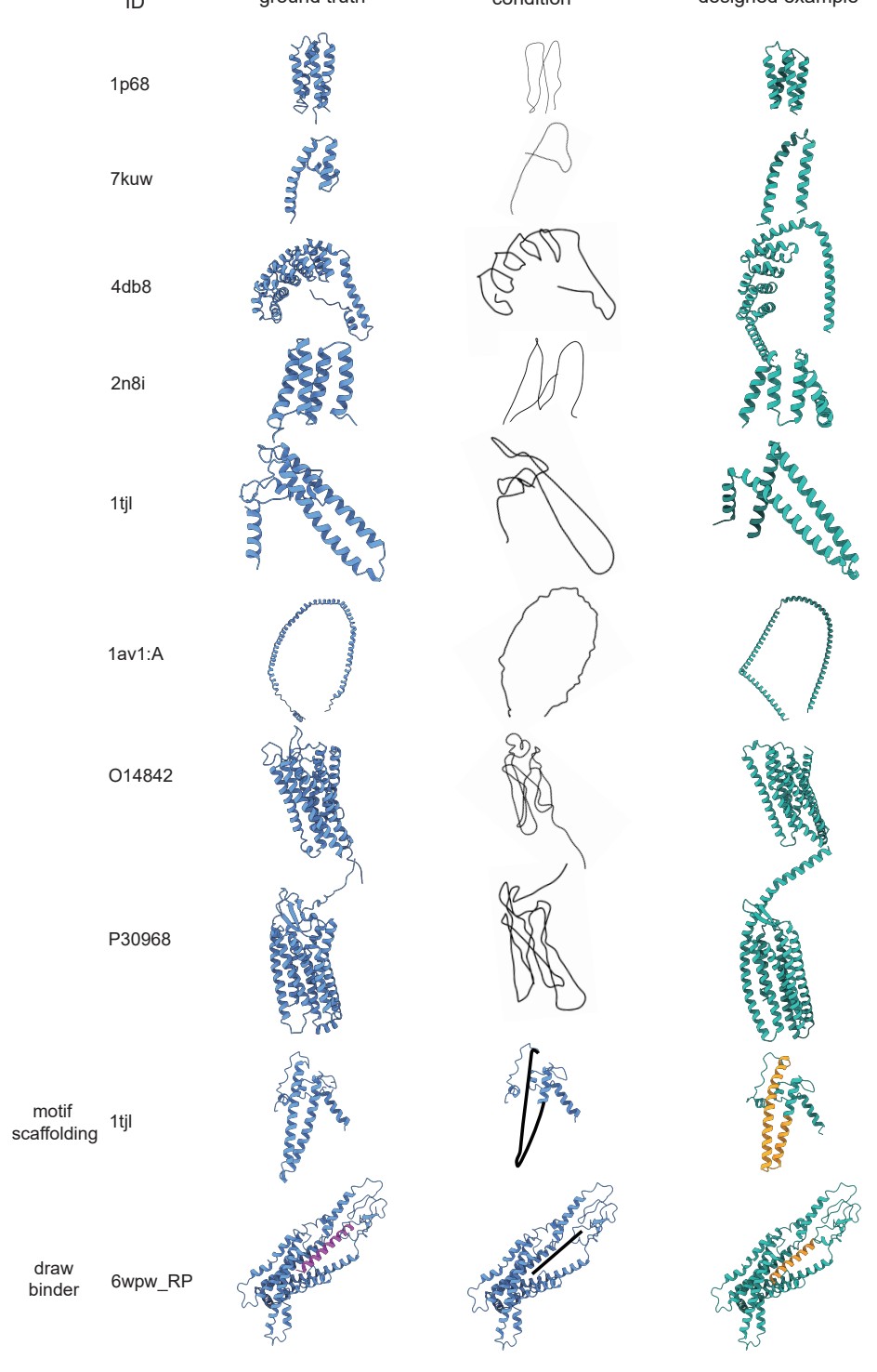

Figure 14: Gallery.

## I  ADDING NOISE TO THE RESTORATION TASK

We simulate the user's drawing process by introducing noise, randomly perturbing each point within a sphere centered at its original position, with a radius (representing noise level) ranging from 0 to 5Å. The results, summarized in Table 9, reveal that fit designability (FD) initially increases before declining. This trend demonstrates the robustness of our method, which can tolerate a certain level of user error while accurately interpreting the intended design. The subsequent decrease in FD, coupled with consistently high levels of confident designability (CD), highlights our method's ability to refine the input condition and generate designable and more reasonable outcomes.

Table 9: Adding noise to the curve condition in Protein Restoration Task.

| Noise Level | HHH_ems | | med | | GPCR | |
|---|---|---|---|---|---|---|
| | CD | FD | CD | FD | CD | FD |
| 0 | 0.832 | **0.654** | 0.870 | 0.734 | **0.936** | **0.792** |
| 1 | 0.768 | 0.602 | **0.924** | **0.802** | 0.902 | 0.694 |
| 2 | **0.882** | 0.540 | 0.668 | 0.428 | 0.510 | 0.326 |
| 3 | 0.802 | 0.414 | 0.432 | 0.290 | 0.104 | 0.062 |
| 4 | 0.714 | 0.428 | 0.232 | 0.102 | 0.188 | 0.126 |
| 5 | 0.784 | 0.392 | 0.286 | 0.166 | 0.298 | 0.192 |

## J  ADDITIONAL EXAMPLE GALLERY

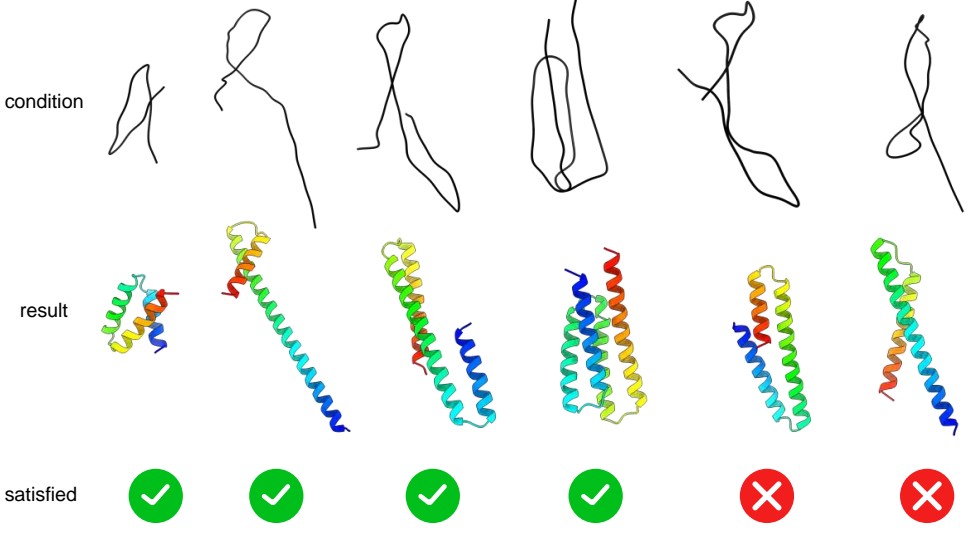

Figure 15: User cases gallery.

## K  BETA SHEET EXAMPLES

We have achieved controlling beta sheet percentage by increasing diversity, inspired by RFDiffusion. The cases are presented in Figure 16. The measures are as follows,

1. partial diffusion (lower self-confidence to increase diversity)
2. increasing lambda and decreasing gamma (lower condition and increase diversity)
3. early stop (fewer diffusion steps to lower the condition and self-confidence)

The cases indicate that it can be extended to beta sheets through sketch features, the same as the helix.

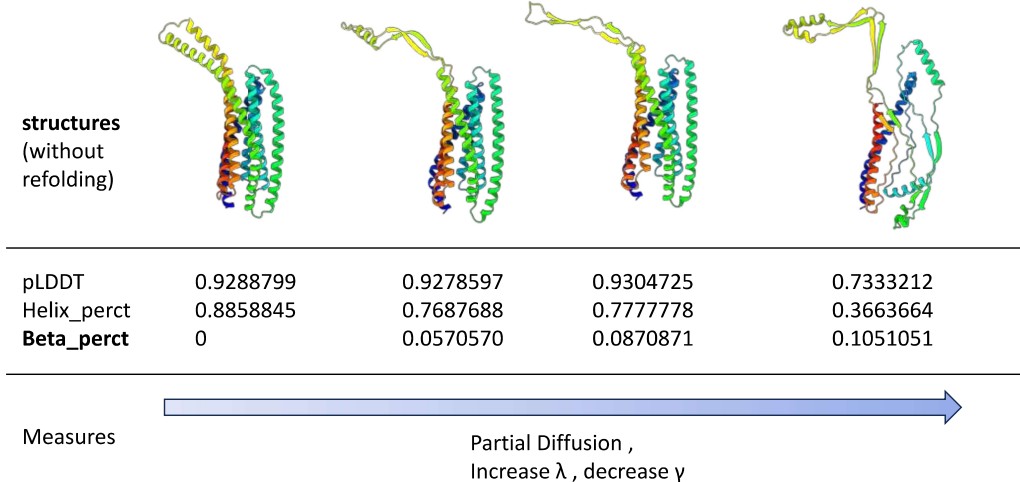

| structures (without refolding) | | | | |
| --- | --- | --- | --- | --- |
| pLDDT | 0.9288799 | 0.9278597 | 0.9304725 | 0.7333212 |
| Helix_perct | 0.8858845 | 0.7687688 | 0.7777778 | 0.3663664 |
| **Beta_perct** | 0 | 0.0570570 | 0.0870871 | 0.1051051 |

Measures

Partial Diffusion ,
Increase λ , decrease γ

Figure 16: Beta sheet examples.

