# OpenReview forum: "ProtPainter: Draw or Drag Protein via Topology-guided Diffusion"
_ICLR.cc/2025/Conference — ICLR 2025 Poster_

### Official Review · Reviewer_VUTJ · 2024-10-31

**Soundness:** 2
**Presentation:** 2
**Contribution:** 2
**Rating:** 6
**Confidence:** 2

**Summary:**

This paper presents a conditional diffusion model that can generate proteins corresponding to 3D topological designs provided by the user.

**Strengths:**

This paper defines a new task, which is a novel attempt, even though I think the significance and novelty of this task may be questionable. The paper introduces CurveEncoder and Helix-Gating for this task, presenting certain innovativeness in its methodology.

**Weaknesses:**

1. The definition of this task may be highly similar to the definition of a problem in chroma [1], although not exactly the same. Here, I first point out where chroma mentions the corresponding task. Figure 3 in the main text of chroma indicates that they have implemented shape conditioning to enable geometric molecular programming. Section R of chroma's supplementary materials, titled "Programmability: Shape," also discusses this. My perspective is that while the objectives of this part of chroma and ProtPainter are not perfectly aligned, the desired effects are very similar. Furthermore, it might be possible to simply adjust chroma's Conditioners to transform chroma's Shape Conditioners into topological Conditioners. My question is: How does ProtPainter's topological conditioning differ from or improve upon Chroma's shape conditioning? Are there specific advantages or capabilities that ProtPainter offers that cannot be achieved by adapting Chroma's approach?


2. A major difference between ProtPainter and chroma's Shape Conditioner is that ProtPainter can additionally specify the proportion of secondary structures, such as the percentage of Helices. However, I think that in many cases when designing the protein scaffold, we want more helical-like secondary structures, as they can improve the thermal stability of the protein. Therefore, specifying the proportion of Helices is unnecessary. Because we don't need low proportion of Helices. It is only when designing motifs or some flexible regions, or under special circumstances, that one would want to avoid Helices and thus need to specify their proportion. But these special circumstances are usually quite complex, and merely specifying topology may be insufficient and unnecessary. Thus, I do not fully understand the significance of this aspect. Of course, I understand that protein design is a very complex issue, and the scenario I mentioned above might only represent one type of design problem. There may be other types of design problems where the user needs to specify the percentage of helices; however, I do not understand the importance of this issue. The paper does not seem to particularly discuss this matter either. Please have the author indicate whether specifying the percentage of helices is important. If it is important, could you explain why and provide applications along with relevant papers?


3. Whether users can provide a correct 3D topological blueprint for successfully generating a protein is questionable. The sketches provided to the diffusion model in this paper are all derived from real proteins. However, in real application scenarios, it is doubtful whether researchers can provide such accurate sketches. If researchers cannot provide very valid sketches, it is uncertain whether ProtPainter can function properly. Moreover, if researchers cannot provide very valid sketches, then chroma's Shape Conditioner might be a more reasonable approach. I suggest that the author provide dedicated experiments to analyze this issue. For example, 1) conduct a user study where a person sketches a protein based on their understanding and professional knowledge of proteins, and then check how successfully ProtPainter can design the protein according to this sketch. 2) In the paper, sketches of proteins are generated from real proteins. During inference, adding noise to the actual sketches to simulate user errors and randomness, see if ProtPainter can still smoothly generate proteins. Additionally, analyze the maximum level of noise that ProtPainter can tolerate.

My understanding may have some inaccuracies, so please feel free to point them out.

[1] Illuminating protein space with a programmable generative model

**Questions:**

See Weaknesses

---

> ### Author Response · Authors · 2024-11-20
> **Response by Authors Part 1/2**
>
> We thank the reviewer for the constructive feedback and the positive remarks on our ideas. We acknowledge that your concerns are mainly about the structural conditioning mechanism and hope our response will resolve your concerns:
>
> **Q1: How does ProtPainter's topological conditioning differ from or improve upon Chroma's shape conditioning? Are there specific advantages or capabilities that ProtPainter offers that cannot be achieved by adapting Chroma's approach?**
>
> While both ProtPainter and Chroma aim to control protein structure, they operate at fundamentally different levels of granularity. Chroma's shape conditioning focuses on overall molecular geometry, or 'volumetric shape', whereas ProtPainter's topological conditioning specifically targets the spatial arrangement and connectivity of secondary structure elements, allowing for direct manipulation of protein topology rather than indirect control through overall shape constraints.
>
> This distinction enables ProtPainter to achieve precise control over backbone topology that would be difficult to accomplish by simply adapting Chroma's shape conditioners. For example, when designing transmembrane protein binders, ProtPainter can explicitly constrain the backbone to maintain proper transmembrane conformations while matching the target surface topology, while Chroma's shape-based conditioning could only fill a given 3D volume without explicit topological control, lead to false conformation that could be impossible to be embedded in the membrane. [1,3,4]
>
> To ensure a fair comparison, we adapted Chroma's shape conditioner to a curve-based approach. Our results in the revised paper Table 3 demonstrate that merely refining the point cloud granularity along curves does not improve FD significantly. Moreover, increasing the sampling radius leads to topological ambiguity due to structural overlapping and decreases the FD. See below:
>
> | **Condition** | **Method**   | **HHH_ems** |           | **med**   |           | **GPCR**  |           |
> | ------------- | ------------ | ----------- | --------- | --------- | --------- | --------- | --------- |
> |               |              | **CD**      | **FD**    | **CD**    | **FD**    | **CD**    | **FD**    |
> | -             | RFDiffusion  | **0.980**       | 0.640     | **0.976** | 0.045     | 0.886     | 0         |
> | SSE           | Chroma       | 0.868       | 0.066     | 0.782     | 0.002     | 0.658     | 0         |
> | PonitCloud    | Chroma r = 0 | 0.812       | 0.088     | 0.464     | 0.044     | 0.208     | 0         |
> |               | r = 1        | 0.784       | 0.144     | 0.544     | 0.042     | 0.182     | 0         |
> |               | r = 2        | 0.862       | 0.026     | 0.640     | 0         | 0.220     | 0         |
> |               | r = 3        | 0.880       | 0.028     | 0.560     | 0.002     | 0.202     | 0         |
> |               | r = 4        | 0.840       | 0         | 0.582     | 0         | 0.166     | 0         |
> | Curve         | ProtPainter  | 0.832       | **0.654** | 0.870     | **0.734** | **0.936** | **0.792** |
>
> Also, I'd like to point out another highlight: ProtPainter's curve-based control enables precise scaffold editing, particularly valuable for multi-state design (See our revised paper section 4.3). Users can generate conformational variants while maintaining designable topological architecture - a capability currently unique among generative models. In a traditional style, the user might use Pymol or ChimeraX to force a whole domain to move, but the scaffolds generated in this way often show low designability and need extra refinement.
>
> In summary, while both structural constraining approaches are valuable, they serve complementary rather than overlapping purposes in the protein design.

---

> ### Author Response · Authors · 2024-11-20
> **Response by Authors Part 2/2**
>
> **Q2: Please have the author indicate whether specifying the percentage of helices is important. If it is important, could you explain why and provide applications along with relevant papers?**
>
> We would like to emphasize that ProtPainter can not only control overall helical proportion but also precisely assign the detailed helical distribution along the protein chain. This capability is crucial for precise protein engineering, particularly in cases where specific dynamic properties are required.
>
> For instance, in membrane proteins, precise control of helical content in transmembrane regions is vital for proper membrane embedding, while maintaining conformational flexibility in other regions, such as intracellular and extracellular domains, is necessary for functions like substrate transport and signal transduction. A prime example is the GPCR Class A protein family [1-GPCR], where the TM5 and TM6 helices are connected by the intracellular loop region ICL3. The length and flexibility of ICL3 enable TM6 movement, which is essential for G protein docking pocket formation and subsequent downstream signaling. Moreover, if you compare the class A GPCRs' proportion of helix part in TM5, you will find that they determine whether the GPCRs are biased to certain downstream signaling (please refer to Figure 5 in the paper [2]).
>
> This represents a natural example of protein dynamic or multi-state protein design, where precise control of both helical and flexible regions enables flexibility controlling and further state switching. Through ProtPainter's capabilities, we aim to implement similar mechanisms in designed proteins, enabling the engineering of such sophisticated conformational dynamics.
>
> **Q3: Moreover, if researchers cannot provide very valid sketches, then chroma's Shape Conditioner might be a more reasonable approach. I suggest that the author provide dedicated experiments to analyze this issue.  For example, 1) conduct a user study where a person sketches a protein based on their understanding and professional knowledge of proteins, and then check how successfully ProtPainter can design the protein according to this sketch.  In the paper, sketches of proteins are generated from real proteins. During inference, adding noise to the actual sketches to simulate user errors and randomness, see if ProtPainter can still smoothly generate proteins. Additionally, analyze the maximum level of noise that ProtPainter can tolerate.**
>
> First of all, in our revised paper Section 4.2, we detailed our methods for users to create valid curves from scratch. As for your suggestion 1), we have planned a simple user study to collect input from expert users:
>
> | Source | scRMSD | scTM  | scTF  | CD   | FD   |
> | ------ | ------ | ----- | ----- | ---- | ---- |
> | Human  | 3.278  | 0.782 | 0.745 | 0.75 | 0.65 |
>
> This experiment collect 20 short proteins generated with the 2D-to-3D method in Section 4.2. We are planning more user studies to comprehensively evaluate the ProtPainter's performance in tasks like binder design or multi-state design.
>
> For suggestion 2), we've conducted an experiment (Supplementary Section I, Table 9) to investigate the sensitivity to noise, see below:
>
> | **HHH_ems** | **HHH_ems** |           | **med**   |           | **GPCR**  |           |
> | ----------- | ----------- | --------- | --------- | --------- | --------- | --------- |
> | noise_level | **CD**      | **FD**    | **CD**    | **FD**    | **CD**    | **FD**    |
> | 0           | 0.832       | **0.654** | 0.870     | 0.734     | **0.936** | **0.792** |
> | 1           | 0.768       | 0.602     | **0.924** | **0.802** | 0.902     | 0.694     |
> | 2           | **0.882**   | 0.540     | 0.668     | 0.428     | 0.510     | 0.326     |
> | 3           | 0.802       | 0.414     | 0.432     | 0.290     | 0.104     | 0.062     |
> | 4           | 0.714       | 0.428     | 0.232     | 0.102     | 0.188     | 0.126     |
> | 5           | 0.784       | 0.392     | 0.286     | 0.166     | 0.298     | 0.192     |
>
> The confident designability (CD) stays consistently high in low noise level (around 1A), suggesting our method can effectively refine slightly imperfect inputs into reasonable structures. The maximum noise tolerance is around 2 for all topology complexity, and when higher than 2A, CD/FD began to drop. Interestingly, we found that small amounts of noise (up to 1-2 Å) can somehow improve fit designability (FD), so we believe the sweet points exist for different topology complexity.
>
> [1] Relevance of G protein‐coupled receptor (GPCR) dynamics for receptor activation, signalling bias and allosteric modulation.
>
> [2] Molecular Determinant Underlying Selective Coupling of Primary G-Protein by Class A GPCRs.

---

> > ### Comment · Reviewer_VUTJ · 2024-11-20
> >
> > Firstly, I would like to express my sincere gratitude to the author for providing the experiments and the efforts made. I have decided to increase my score. However, I still believe that this paper is on the borderline between acceptance and rejection. 1. I do not have additional concerns about the first two issues and hope that the author will further refine the writing in the next version. 2. Regarding the noise level experiment, the results show that the robustness of this model is not very good, with a maximum tolerance of only 2A noise. I believe that the noise in the curves drawn by experts is greater than 2A. Perhaps the author should consider adding certain random perturbations to the curves during training as a form of data augmentation to enhance the model's robustness. However, considering the experiments already provided by the author and the high probability that this issue can be resolved, I do not think it is a serious problem; it is just that the current work has not been fully refined. 3. I believe that this model has some potential in addressing the specific protein design problems mentioned by the author; however, this article is merely an attempt, and there is still a considerable distance from practical application. Therefore, while this article makes a certain contribution, it is not a particularly significant one. Finally, I will lower my confidence score. Due to my limited knowledge of biology, although I can understand and acknowledge the author's ideas regarding related design issues, because of the limitations of my knowledge, I may still reach incorrect conclusions.

---

> ### Author Response · Authors · 2024-11-21
> **Response by Authors**
>
> Thank you for your active response and for increasing the rating score.
>
> **1. About robustness**
>
> Users typically draw curves on a painting board or a software platform such as ChimeraX, where the scale is in tens of centimeters. These curves are then proportionally scaled down to Angstroms (A), at which point noise is introduced. So the tolerance of user-drawn curves is significantly greater than 2A, depending on the specific drawing tool used. To improve robustness, we have already incorporated techniques such as re-sampling curves and masking partial labels during the training of our CurveEncoder (as described in Appendix D.4). As you suggest, it is a good data augmentation method by adding certain random perturbations. We will try it later.
>
> **2. About Further application**
>
> Our work is an attempt to bridge the gap between general users and complex protein design, which has not been explored before. As you said, it has potential in problems like multi-state design and binder design. It has already demonstrated success in generating designable proteins for desired tasks, hinge[1] for example. It reduces the steps and time required for researchers. Of course, the results need further wet laboratory experimental studies.
>
> We are happy to communicate and share our diffusion method in protein design tasks with you. Your advice about the experiments is valuable. And we will keep on polishing and refining our paper.
>
> [1] Design of stimulus-responsive two-state hinge proteins

---

### Official Review · Reviewer_Rep2 · 2024-11-02

**Soundness:** 2
**Presentation:** 2
**Contribution:** 2
**Rating:** 5
**Confidence:** 4

**Summary:**

This paper tackles controllable protein structure generation. Specifically, it proposes a method to extract an editable "curve representation" that describes the chain together with some secondary structure annotations from protein backbones. Given that curve, new proteins with a similar topology can be synthesized by conditioning a protein backbone generator on the curve. Modifying the curve representation enables to edit the protein structure in a controllable manner. The curve representation is extracted from given proteins using a CurveEncoder module. The method is validated on protein restoration tasks where the goal is to reconstruct proteins with similar topology as the input proteins from which the curve representation was extracted. Additionally, some editing tasks are shown that topologically and spatially modify given proteins.

**Strengths:**

**Originality**: The proposed method is novel, to the best of my knowledge, building on the conditioning framework from Choi et al. (ILVR). Some related works exist, however, and the method is quite related to DiffTopo.

**Experiments**: The paper successfully demonstrates that the curve representation can encode spatial protein layout and topology and that it can be used to recreate proteins of similar topologies. The qualitative protein editing experiments are also interesting.

**Clarity**: While it is overall possible to generally follow the paper, the paper writing is of poor quality and some things remain somewhat unclear (see weaknesses below).

**Quality and Significance**: Due to the paper's weaknesses (see below), I think the significance and quality of the paper is low.

**Weaknesses:**

I see several weaknesses:

- The method can only generate alpha helices or coil/loop regions, but not beta sheets. This is a major weakness and severely reduces the method's broad applicability and significance. This is also not discussed in detail and it remains unclear why exactly that is the case. Why can beta sheets not be treated similarly like alpha helices?
- The only experiment where ProtPainter is compared to existing baselines is in Table 3. However, I think these results are misleading. These baseline models (RFDiffusion, Chroma, and ProteinGenerator) are not conditioned on topological, spatial layouts of the proteins, it seems, as is the case for ProtPainter. Hence, we cannot expect those models to spatially and topologically recreate any given proteins and it is not surprising that those baselines achieve low scTF/FD scores (and for designability [CD], ProtPainter isn't actually outperforming the baselines in all cases). In other words, the baselines are not chosen appropriately.
  - Chroma has a method to constrain protein generation on shapes. I think it could be appropriate to extract shapes from the input proteins (analogous to the curves) and condition Chroma on those, if possible.
  - I also think comparisons to TopoDiff and/or DiffTopo would be appropriate, if possible. They may not offer fine editing control, but these methods can also encode protein topologies and spatially reconstruct, I think. These works are actually discussed in the paper and are arguably most related.
- The designability definition used in this paper relies on pLDDT > 70 and scTM > 0.5. However, most works in the literature these days use a stricter designability definition of scRMSD < 2 Angstrom. I would suggest the authors to use this instead.
- In Section 3.2, the paper writes that ProtPainter leverages RoseTTAFold to predict $\hat{z}_0$ (also in figure 1). I would have thought that the authors use the diffusion model RFDiffusion (which is initialized from RoseTTAFold, but itself is not a folding model anymore after training). This also isn't well explained in the very short "Train" section in the appendix. It is not entirely clear how the model is trained.
- The model only shows protein restoration and protein editing tasks, but no de novo generation from scratch. It would be great if the authors could also demonstrate the generation of designable proteins from curves drawn entirely from scratch or generated by another curve generator (the very related TopoDiff and DiffTopo do such things, too).
- As mentioned previously, the writing quality is generally low, with various grammar errors. This negatively affects the quality of the paper.
- The ILVR method is cited, but without looking at the Choi paper the ProtPainter method is hard to follow in detail. More details would be helpful and a more critical discussion and motivation of the chosen frame filter operation would be useful, among other things, for example.

In conclusion, while ProtPainter's curve representation is interesting and the protein editing capabilities could be useful, due to the above weaknesses I do not think that the paper meets the bar for publication.

**Questions:**

I do not have any further questions to the authors.

---

> ### Author Response · Authors · 2024-11-20
> **Response by Authors Part 1/3**
>
> We sincerely thank the reviewer for your interest in our curve-based editing approach and constructive feedback. Your main concerns are about its applicability, benchmarking, and representation. We would like to address them to our knowledge.
>
> **Q1: The method can only generate alpha helices or coil/loop regions, but not beta sheets. This is a major weakness and severely reduces the method's broad applicability and significance. This is also not discussed in detail and it remains unclear why exactly that is the case. Why can beta sheets not be treated similarly like alpha helices?**
>
> Actually, the current limitations to alpha helices and loops are based on practical considerations:
>
> 1. Our curve-based method was specifically designed for **precise control over protein topology**, which particularly aligns with practical applications in binder design and multi-state protein design. In these scenarios, alpha-helical scaffolds are often preferred due to their superior thermal stability and higher designability. For instance, widely applied binder protocols like Cao's protocol [1] specifically recommend triple helical scaffolds (HHH), and successful multi-state designs [2,3,4] predominantly utilize helical components for their reliability and beta-sheet specification is not a major concern.
> 2. While beta sheets are crucial structural elements, controlling their generation presents unique challenges that may not be optimally addressed through curve-based methods. Beta sheets depend on complex networks of non-adjacent residue interactions and specific hydrogen bonding patterns. User-specification through curves might struggle to locally edit such structures. We believe coarse-grained spatial constraints (like Chroma) might be more effective than precise curve guidance, as they better consider the overall spatial relationships essential for sheet formation.
>
> Nevertheless, we agree this is a useful direction for improving ProtPainter's broad applicability. We believe beta-sheet can be treated in the exact way like alpha-helix, (i.e., sketching, followed by a sketch-guided diffusion). However, we are suggesting a hybrid way enabling both delicate curve-based control on helices, and point-cloud-based spatial brush for sheets to specify the overall structure could be a better implementation in future development, as they require different granularity.
>
>
>
> **Q2: The baselines are not chosen appropriately. Chroma has a method to constrain protein generation on shapes. I think it could be appropriate to extract shapes from the input proteins (analogous to the curves) and condition Chroma on those, if possible. I also think comparisons to TopoDiff and/or DiffTopo would be appropriate, if possible.**
>
> The comparisons to SSE-conditioned methods are conducted to prominent our fine-grained and more precise constraint, not to outperform them on FD. In our revised paper Table 3, we've added the comparisons to Chroma (point-cloud conditioned) and TopoDiff (latent mediated), which are more relevant:
>
> 1. For Chroma shape conditioning, we implemented a curve-to-point-cloud converter, which can generate a cylinder point cloud with different radii around the given curve. Results suggest that through this shape-based method, Chroma is able to generate backbones with high CD but couldn't effectively restore the proteins (low TF), even when low radius of 1A. We reason that the point-cloud method causes topology ambiguity.
> 2. For TopoDiff, we've designed a method to adapt it from an unconditional latent diffusion model to a curve-guided diffusion. Please kindly refer to the Section Baselines. Results indicate that sketch-mediated latent diffusion cannot capture the user-drawn topology and thus get low scTF scores.
> 3. The explanation for not including DiffTopo comparison: We agree that our paper shares a common idea with DiffTopo that sketch-mediated guidance to Diffusion is an effective structural conditioning mechanism. But DiffTopo actually doesn't implement a user-specified structural conditioning mechanism and might not be a proper baseline for **protein restoration**. As described in their paper [5] Section A.1.4, DiffTopo generates the coarse-grained (CG) topology representation (which in our paper is the SSE-labelled curves drawn by user). Based on these CGs, sketches are parametrically generated and are applied external partial diffusion with RFDiffusion followed by a hallucination optimizing protocol. In a word, DiffTopo can be viewed as **an SSE-conditioned sketch generator**, which is not suitable to be a baseline.

---

> ### Author Response · Authors · 2024-11-20
> **Response by Authors Part 2/3**
>
> **Q3: The designability definition used in this paper relies on pLDDT > 70 and scTM > 0.5. However, most works in the literature these days use a stricter designability definition of scRMSD < 2 Angstrom. I would suggest the authors to use this instead.**
>
> This allows us to clarify our metric choice, which aligns with our fundamental objective of evaluating overall topology restoration.
>
> While scRMSD<2Å is indeed a valuable and stringent metric widely used in the field, our focus on topology restoration led us to select scTM>0.5 as our primary metric. As demonstrated in previous work ([6] Lin & Alquraishi, 2023), scTM>0.5 effectively indicates whether generated and designed structures share the same fold topology, which is precisely what we aim to evaluate.
> To illustrate the implications of both metrics, we present comparative results:
> Table with scRMSD<2Å, plddt> 70 as the CD/FD threshold:
>
> | **Condition** | **Method**  | **HHH_ems** |           | **med**  |           | **GPCR** |           |
> | ------------- | ----------- | ----------- | --------- | -------- | --------- | -------- | --------- |
> |               |             | **CD**      | **FD**    | **CD**   | **FD**    | **CD**   | **FD**    |
> | -             | RFDiffusion | **0.96**    | **0.604**     | **0.94** | 0.1       | **0.78** | 0         |
> | SSE           | Chroma      | 0.708       | 0.056     | 0.662    | 0         | 0.368    | 0         |
> | PonitCloud    | Chroma r=0  | 0.632       | 0.104     | 0.254    | 0.024     | 0.008    | 0         |
> |               | r = 1       | 0.564       | 0.082     | 0.232    | 0.012     | 0.012    | 0         |
> |               | r = 2       | 0.568       | 0         | 0.242    | 0         | 0.024    | 0         |
> |               | r = 3       | 0.702       | 0.022     | 0.402    | 0.002     | 0.020    | 0         |
> |               | r = 4       | 0.642       | 0         | 0.342    | 0         | 0.010    | 0         |
> | Topology      | TopoDiff    | 0.936       | 0.162     | 0.824    | 0.028     | 0.018    | 0         |
> | Curve         | ProtPainter | 0.604       | 0.596 | 0.384    | **0.274** | 0.106    | **0.078** |
>
> The scRMSD results show a sharp performance drop as topology complexity increases, reflecting its sensitivity to small structural differences and longer sequences. While this sensitivity is valuable for many applications, we find that scTM better serves our specific goal of **assessing overall fold topology restoration**. This is further supported by our analysis in Supplementary A.3, which demonstrates how scTM captures topological similarity even when local structural details vary. Therefore, we kindly suggest maintaining our definition of CD/FD to scTM as it better aligns with our purpose of controlling 'overall topology'.
>
> However, to provide a complete picture, we are happy to expand Table 3 to include scTM and scRMSD. This would allow readers to evaluate performance from both perspectives - the stringent local structural accuracy (scRMSD) and the overall topological similarity (scTM) - while maintaining our CD/FD definition primarily to focus on topology restoration.
>
> **Q4: In Section 3.2, the paper writes that ProtPainter leverages RoseTTAFold to predict $z^0$ (also in figure 1). I would have thought that the authors use the diffusion model RFDiffusion (which is initialized from RoseTTAFold, but itself is not a folding model anymore after training). This also isn't well explained in the very short "Train" section in the appendix. It is not entirely clear how the model is trained.**
>
> We would like to clarify this part in detail:
>
> You are absolutely correct that RFDiffusion is initialized from RoseTTAFold. However, a major difference between RFDiffusion and other diffusion models is that it doesn't directly predict the $x_{t-1}$ from $x_t$, but first estimates T=0 structure $\hat{x}^0_t$ (containing
> $\hat{z}^0$) from $x_t$ and then interpolate it with $x_t$ to get $x_{t-1}$ in each timestep. The fine-tuned version of RoseTTAFold serves as a key component specifically to estimate the T=0 structure (translation part $\hat{z}^0$, or full frame $\hat{x}^0_t$) at each diffusion timestep.
>
> Inspired by this, ProtPainter implements a **training-free**, guided-sampling approach based on RFDiffusion. Rather than fine-tuning or retraining the base diffusion model, we use interpolation between $z_t$ and scaled y(yt-1) during the sampling process.
>
> In our paper, we provide theoretical and experimental validation for this training-free approach. It offers several advantages such as,
>
> - No need for costly diffusion model retraining or finetuning, for more efficient training of CurveEncoder can work well.
> - It is actually more interpretable and controllable to condition on translation (Cα coordinates) than latent space conditioning.
> - It can be implemented as a plug-and-play solution with various base models.

---

> ### Author Response · Authors · 2024-11-20
> **Response by Authors Part 3/3**
>
> **Q5: The model only shows protein restoration and protein editing tasks, but no de novo generation from scratch. It would be great if the authors could also demonstrate the generation of designable proteins from curves drawn entirely from scratch or generated by another curve generator (the very related TopoDiff and DiffTopo do such things, too).**
>
> We add **three more** approaches to generate curves from scratch, described in the revised version's Section 4.2. Our second method is that we implement a simple algorithm to transform a 2D curve to 3D by assigning random, smooth depth to it. This enables users to create a reasonable curve from scratch. An additional 2D curve generator is implemented to randomly create short 2D curves with 3-5 turns, enabling unconditional-generated 3D curves. Our results in the revised paper Table 4 evaluates their performance in short proteins (35-100aa), where 80 2D curves are generated randomly and 20 are drawn by users:
>
> | Source          | scRMSD | scTM  | scTF  | CD     | FD    |
> | --------------- | ------ | ----- | ----- | ------ | ----- |
> | Curve Generator | 3.565  | 0.768 | 0.696 | 0.8875 | 0.575 |
> | User            | 3.278  | 0.782 | 0.745 | 0.75   | 0.65  |
> | All             | 3.508  | 0.771 | 0.706 | 0.86   | 0.59  |
>
> The results indicate the potential of the curve generator to help create confident designable backbones for short proteins while retaining a reasonable FD. If you are satisfied with this de novo generation implementation, we are happy to test it for longer proteins.
>
> **Q6: As mentioned previously, the writing quality is generally low, with various grammar errors. This negatively affects the quality of the paper.**
>
> In the revised version of the paper, we have thoroughly addressed the grammatical errors and reorganized the writing to enhance readability and fluency. Should you notice any remaining issues, we would be grateful if you could kindly point them out so we can make further improvements. Thank you for your understanding and support.
>
> **Q7: The ILVR method is cited, but without looking at the Choi paper the ProtPainter method is hard to follow in detail. More details would be helpful and a more critical discussion and motivation of the chosen frame filter operation would be useful, among other things, for example.**
>
>
> To conclude, the design of the frame filter operation is inspired by the ILVR[7] approach, where a low-pass filtering operation is used to align low-dimensional features in a super-resolution imaging task. We adapt this idea to the conditional protein design setting, introducing key variations to accommodate the dimensional differences between protein generations and reference curves. Below, we elaborate on the details of the motivations and relations:
>
> In ILVR, the generation $x$ is refined conditioned on a reference $y$ through:
>
> \begin{equation}
>     p_\theta(x_{t-1} | x_t, y) \approx p_\theta(x_{t-1} | x_t, \phi(x_{t-1}) = \phi(y_{t-1})).
> \end{equation}
>
> where $\phi$ represents a linear low-pass filtering operation that ensures the low-frequency features of the reference and the generation remain consistent.
>
> In our work, we draw an analogy between super-resolution tasks in imaging and conditional protein design. Specifically, the input curve condition serves as the reference image $y$, while the generated protein backbones correspond to the generative image $x$. However, unlike images, protein structures and curves differ in dimensionality, making direct alignment through traditional upsampling or downsampling impractical.
>
> To address this challenge, we propose a $\textbf{Curve Encoder}$ that upsamples the input condition curve into a higher-dimensional "sketch." Concurrently, we apply a frame filter operation $\phi$ to transform the generated backbones into a "frame." Both the sketch and frame are designed to reside in the same dimensional space, effectively representing $C_\alpha$ transitions. This alignment facilitates straightforward guidance of the backbone frame by the reference sketch, enabling precise generation.
>
> We hope this explanation, along with the revised introduction and method sections that are highlighted in red in the updated paper, clarifies the design and motivation of our method.
>
> [1] Design of protein-binding proteins from the target structure alone.
>
> [2] Design of stimulus-responsive two-state hinge proteins.
>
> [3] De novo design of allosterically switchable protein assemblies.
>
> [4] Design of facilitated dissociation enables control over cytokine signaling duration.
>
> [5] DiffTopo: Fold Exploration Using Coarse Grained Protein Topology Representations.
>
> [6] Generating Novel, Designable, and Diverse Protein Structures by Equivariantly Diffusing Oriented Residue Clouds.
>
> [7] ILVR: Conditioning Method for Denoising Diffusion Probabilistic Models

---

> ### Comment · Reviewer_Rep2 · 2024-11-25
> **Response to Rebuttal**
>
> I would like to thank the authors for their detailed reply to my review and for their efforts in addressing my questions. Overall, my general conclusions remain.
>
> - I agree that for overall topological reconstruction scTM>0.5 seems like a reasonable metric. But for *designability* (i.e. whether there exists an amino acid sequence that accurately fold into the structure of question) I still believe a stricter criterion to be appropriate, i.e. the widely used scRMSD<2A. Topological reconstruction accuracy and designability are two separate questions -- this is not appropriately discussed and distinguished. Unfortunately, the scRMSD of ProtPainter (Tables 3 and 4) seems to be generally >2A, indicating poor designability (on this note, it could be good to also report the percentage of designable structures, as is common practice in the field).
> - While it is true that only alpha helix-based protein design has various applications, it remains problematic that beta sheets cannot be tackled with this method. This limits the broad applicability of the method and suggests that future work will not build on this method, but instead develop new methods that are more flexible to also handle beta sheets.
> - I still think the clarity of the paper could be better. For instance, the explanation of RFDiffusion vs. RosettaFold still seems confusing. RosettaFold is a folding model that takes an amino acid sequence as input, but my understanding is that ProtPainter uses the $x_0$ prediction from RFDiffusion, which does *not* have a sequence input. If that is the case, then the authors should clearly write that they use the $x_0$ prediction from RFDiffusion and *not* the amino acid-conditioned RosettaFold.
> - I appreciate the additional experiments on de-novo generation. However, the performance does not seem high. In particular the high scRMSD is concerning (see above). Moreover, unfortunately no samples are visualized for the de-novo generation experiments. This would help to get intuitions about how flexible the model is with respect to different manually specified curves.

---

> ### Author Response · Authors · 2024-11-29
> **Response by Authors**
>
> Thank you for your response. We would like to address your points:
>
> **About designability**
>
> Regarding the scRMSD threshold, while we understand that scRMSD<2Å is commonly used for designability assessment, we believe this criterion may be overly strict for our specific use case. Our method aims to provide a novel approach for protein backbone generation guided by curves, where topological accuracy is the primary goal.
>
> **About beta sheet**
>
> We still hold the belief that alpha helix is enough for topology control jobs like binder design and multi-state protein design. And our method is proposed to solve it.
>
> Considering broad applicability as you mentioned, we have achieved controlling beta sheet percentage by increasing diversity, inspired by RFDiffusion. The cases are presented on [Anonymized Repository - Anonymous GitHub](https://anonymous.4open.science/r/Appendix_fig-F249/Beta.md). The measures are as follows,
>
> 1. partial diffusion (lower self-confidence to increase diversity)
> 2. increasing lambda and decreasing gamma (lower condition and increase diversity)
> 3. early stop (fewer diffusion steps to lower the condition and self-confidence)
>
> The cases indicate that it can be extended to beta sheets through sketch features, the same as the helix. But it may take time to conduct an experiment and we don't target this topic. Hope these cases can answer your concern about generalization.
>
> **About the clarity of RFDiffusion versus RosettaFold usage**
>
> To clarify, we utilize the x0 prediction from RossetaFold with masked sequence input, inspired by RFDiffusion. We color the particular part brown in the Method Section. If you are still confused about this part, please look up the diagram that we have posted here [Anonymized Repository - Anonymous GitHub](https://anonymous.4open.science/r/Appendix_fig-F249/Diagram.md).
>
> **About visualization**
>
> We've added representative examples of de novo generation in **Appendix J** to better illustrate the model's capabilities and limitations in following different curve specifications. While the scRMSD values are higher than the traditional designability threshold, we believe the results demonstrate the feasibility of curve-guided backbone generation, opening up new possibilities in protein design.

---

> > ### Comment · Reviewer_Rep2 · 2024-11-30
> > **Response**
> >
> > I would like to thank the authors for their follow-up message. Overall, I appreciate the efforts by the authors in adding additional experiments and explanations during the rebuttal. Therefore, I am willing to raise my score. However, my overall conclusions and skepticisms about the work remain.

---

> ### Author Response · Authors · 2024-11-30
> **Response by Authors**
>
> Thank you for feedback and increasing the score! We are glad our responses addressed your concerns. Your advice is valuable and helps to refine our work.

---

### Official Review · Reviewer_WjYS · 2024-11-04

**Soundness:** 3
**Presentation:** 3
**Contribution:** 3
**Rating:** 8
**Confidence:** 5

**Summary:**

This work proposes ProtPainter, a diffusion-based method for generating protein backbones with topological constraints. The proposed CurveEncoder enables sketching, bridging the gap between a conditional curve and backbone modality. The sketch guided sampling based on Helix-Gating scheduling enables efficient feature fusion and retains the most generation ability of RFDiffusion. The method enables powerful topology space navigation through curve-based operations, enriching current protein design methods.

**Strengths:**

1.	The author provides a novel and meaningful task as Curve and topology-based protein design, which offers relatively precise and flexible global topology control. The point is the authors connect curves and SSE, which serves as a blueprint and a bridge between human-draw, maybe not so physically reasonable curves and designable protein backbones
2.	The paper proposes a simple and elegant method for curve-guided protein backbone sampling, including curve sketching and Sketch-guided backbone sampling
3.	This paper presents a simple, training-free sketch-guided backbone sampling method based on interpolation between zt and scaled y (yt-1), without the need for expensive retraining of large protein structure generative models.
4.	The author proposes Helix-Gating to improve the training-free guided sampling performance further.

**Weaknesses:**

1.	The proposed method underperforms when being conditioned on complex interlocking topologies, and this framework has not supported SSE annotation with beta sheets or others so far.
2.	For binder design, ProtPainter just provides an empirical conformation estimation. Further optimization and validation are required.
3.	For users facing real-world tasks, a more user-friendly, responsive interface that allows users to draw or drag proteins with ease.

**Questions:**

1.	The main experiments are based on Protein Restoration task. Is it difficult for a general user to draw a reasonable curve?
2.	To draw a good curve requires both drawing skills and knowledge about protein topology, if the drawn curve is not so good (for example, the drawn global topology is not physically stable), will the generation performance drop?
3.	Do the authors have some ideas about correcting the draw error mentioned in the the Q2?

---

> ### Author Response · Authors · 2024-11-20
>
> We sincerely thank the reviewer for your thorough, constructive feedback and the recognition of our contributions to bridging human-drawn curves with protein backbone generation! We would like to address your concerns regarding user interaction and curve drawing reliability (our updated manuscript can be downloaded at the top of this page):
>
>  **Q1: The main experiments are based on Protein Restoration task. Is it difficult for a general user to draw a reasonable curve?**
>
> We acknowledge that a general user might have difficulty directly drawing a 3D curve (a VR controller might be ideal!). Our revised paper Section 4.2 describes three approaches for users to create reasonable curves at ease:
>
> 1. Users can start with a predefined curve (from real proteins and our curated curve libraries) and drag anchor points to reshape it. This is our recommended way for users to create a new curve.
> 2. Users can first draw a 2D curve, which is assigned random and smooth depths then and guides the generation from scratch. Our additional user study in Table 4 indicates the effectiveness of this method.
> 3. For binder design, the user can draw a curve to match the surface.
>
> We found that given the three approaches above, even users with basic protein knowledge can create reasonable short backbones after minimal training. We've conducted a basic user study with method 2 (2D-to-3D) to evaluate the drawing performance. The study collected 20 user-drawn toy scaffolds (35-100aa) from structure biologists, and they demonstrate reasonable CD/FD:
>
> | Source | scRMSD | scTM  | scTF  | CD   | FD   |
> | ------ | ------ | ----- | ----- | ---- | ---- |
> | User   | 3.278  | 0.782 | 0.745 | 0.75 | 0.65 |
>
> **Q2: To draw a good curve requires both drawing skills and knowledge about protein topology, if the drawn curve is not so good (for example, the drawn global topology is not physically stable), will the generation performance drop?**
>
> We first studied this through a noise perturbation experiment (shown in Supplementary Section I, Table 9) to imitate the user input, where we added random Gaussian variations to the curve points. The confident designability (CD) and fit designability (FD) stay high in low noise levels (around 1-2A) across topology complexity, suggesting our method can effectively refine slightly imperfect inputs into reasonable structures. The maximum noise tolerance is around 2A for all topology complexity, and when higher than 2A, CD/FD all began to drop for medium and hard topology (GPCR). Interestingly, we found that small amounts of noise (up to 1-2 Å) can actually improve fit designability (FD) for column med, which means a sweet point might exist for different types of topology.
>
> |             | **HHH_ems** |           | **med**   |           | **GPCR**  |           |
> | ----------- | ----------- | --------- | --------- | --------- | --------- | --------- |
> | noise_level | **CD**      | **FD**    | **CD**    | **FD**    | **CD**    | **FD**    |
> | 0           | 0.832       | **0.654** | 0.870     | 0.734     | **0.936** | **0.792** |
> | 1           | 0.768       | 0.602     | **0.924** | **0.802** | 0.902     | 0.694     |
> | 2           | **0.882**   | 0.540     | 0.668     | 0.428     | 0.510     | 0.326     |
> | 3           | 0.802       | 0.414     | 0.432     | 0.290     | 0.104     | 0.062     |
> | 4           | 0.714       | 0.428     | 0.232     | 0.102     | 0.188     | 0.126     |
> | 5           | 0.784       | 0.392     | 0.286     | 0.166     | 0.298     | 0.192     |
>
> Besides random noise variations, the errors in overall topology (severe clash, interlock) would lower the topology fitness, but the designability (CD) will not be severely interfered, as suggested in Table 3. An interesting fact is that although a curve might empirically seem unstable, it could be designable. Take the "L"-shaped topology in scaffold (3) of Figure 4 for example, while it might look unstable at first glance, it actually achieves high designability scores.
>
> **Q3: Do the authors have some ideas about correcting the draw error mentioned in the Q2?**
>
> Currently, we have taken some measures to lower the model sensitivity to curve quality.
>
> 1. The raw curves are re-sampled and smoothed with a 3D spline function to generate regular curves. It is mentioned in Appendix A. and it should handle some common drawing imperfections.
> 2. When training the CurveEncoder, we randomly mask the curve to enhance the robustness of the SSE assignment.
>
> Also, as you mentioned, metrics about the topology errors (severe clash, interlock, etc) would be beneficial for the user to instantly know whether their curves are physically stable. For such metrics, we plan to implement them in the front-end logic. For example: assigning a collision radius for each point.

---

### Author Response · Authors · 2024-11-20
**Overall Response by authors**

We thank all reviewers for their constructive feedback and the time taken to review!



# Main Changes

During the rebuttal, we received lots of valuable feedbacks and concerns, which are mainly about generalization, practicality, comparison, and method demonstration (related to ILVR and RFDiffusion). We have addressed almost all of them and updated the manuscript which can be downloaded above with new results and writing improvements. The main additions are highlighted in blue (the final version will not contain the coloring). The main changes include:

- Section 3.2 with clearer clarification of the relation with ILVR[1] and the motivation of the frame filter, as suggested by reviewer **Rep2** (highlighted in red).
- Section 4.1
  - A new version of Table 3 with
    - additional comparison to Chroma[2] and TopoDiff[3], as suggested by reviewer **Rep2** and **VUTJ**. They are more relevant baselines that are conditioned on point cloud or topology latent. Results indicate that ProtPainter remains more competitive in understanding overall topology, with higher scTF and FD.
    - additional metrics (scRMSD, scTM, scTF) for the restoration task considering a suggestion from reviewer **Rep2**.
  - Rearrangement of Metrics and Comparison.
- A new Section 4.2 USER STUDY: CURVE FROM SCRATCH with
  - three approaches to generate curves from scratch.
  - a simple user study to empirically evaluate the controllability of our curve generation method 2.
- A new Appendix Section I with an experiment to investigate the sensitivity to curve noise, suggested by reviewer **VUTJ**.
- A new Appendix Section J with examples to show our capability and weakness facing de novo design, suggested by reviewer **Rep2**.
- Miscellaneous
  - Section 3.4 with more simple clarification.
  - Simplification of the Section 5 CONCLUSION
  - Some fixes of grammar errors and writing misunderstandings, suggested by reviewer **Rep2**.
  - Some fixes of details in figures.

To add in the future (camera-ready version):
- The capability to control beta sheets, shown in [Anonymized Repository - Anonymous GitHub](https://anonymous.4open.science/r/Appendix_fig-F249/Diagram.md), suggested by reviewer **Rep2**.

# Reclaim
Our **contribution** is bridging the gap between user will and protein generation, enabling more precise control and flexible applications. The main **novelty** lies in our structural condition, user-defined curve, which is more accurate than SSE, shape (point cloud), topology latent, and so on. And we propose a benchmark to prove it. To implement the conditional generation, the condition is fused into diffusion process on the frame dimension, inspired by ILVR [1]. The additional experiments demonstrate its practicality by users and extension to beta sheets.

[1] ILVR: Conditioning Method for Denoising Diffusion Probabilistic Models

[2] Illuminating protein space with a programmable generative model

[3] Improving diffusion-based protein backbone generation with global-geometry-aware latent encoding

---

### Meta-Review · Area_Chair_8raZ · 2024-12-20

**Metareview:**

The paper introduces ProtPainter, a diffusion-based method for generating protein backbones from 3D curves. It uses CurveEncoder for sketch generation and DDPM for backbone synthesis, with Helix-Gating for scaling control. A new benchmark and metric (scTF) for topology-conditioned protein generation is proposed. Experiments show ProtPainter generates topology-consistent (scTF > 0.8) and designable (scTM > 0.5) backbones, demonstrating flexibility across tasks.

**Strengths:**

1. The authors present a novel and meaningful task focused on curve- and topology-based protein design.

2. To address this challenge, the authors introduce CurveEncoder and Helix-Gating, offering innovative contributions to the methodology.

3. The paper effectively demonstrates that the curve representation can capture both spatial protein layout and topology, and that it can be used to reconstruct proteins with similar topologies.

**Weaknesses:**

1. The method currently supports the generation of alpha helices and coil/loop regions, but does not accommodate beta sheets.

2. Some of the metrics used in the paper require further clarification.

**Overall:**

This paper presents an innovative and promising approach to protein design. The proposed methods show sufficient novelty, and the experiments provide strong support for their effectiveness. While there are some limitations, this work represents a valuable first step and can facilitate future research in protein design.

**Additional Comments On Reviewer Discussion:**

During the rebuttal, the authors thoughtfully addressed the reviewers' feedback and concerns, which mainly focused on generalization, practicality, comparison, and method demonstration (particularly regarding ILVR and RFDiffusion). They have resolved almost all issues, updating the manuscript with new results and improved writing.

The remaining major concerns are raised by Reviewer Rep2, who has increased the score but still expresses some reservations. The first concern is that the approach currently only works for alpha helices. The second relates to the choice of certain metrics used by the authors. While Reviewer Rep2 questions the overall significance of the work, he acknowledges the novelty of the method and the improvements made during the rebuttal. He is on the borderline but would not object to the paper’s acceptance.

Considering all reviewers' comments during the rebuttal period, I believe the strengths of the paper outweigh the weaknesses, and I therefore recommend acceptance.

---

### Decision · Program_Chairs · 2025-01-22

Accept (Poster)